# PIANO: PHYSICS INFORMED AUTOREGRESSIVE NETWORK

## ABSTRACT

Accurately solving time-dependent partial differential equations (PDEs) is fundamental to science and engineering. Physics-Informed Neural Networks (PINNs) solve PDEs using deep learning. However, PINNs perform pointwise predictions that neglect the autoregressive property of dynamical systems, leading to instabilities and inaccurate predictions. We introduce Physics-Informed Autoregressive Networks (PIANO)—a framework that redesigns PINNs to model dynamical systems. PIANO operates autoregressively, explicitly conditioning future predictions on the past. It is trained with self-supervised rollouts that enforce physical consistency. We present a rigorous theoretical analysis demonstrating that PINNs suffer from temporal instability, while PIANO achieves stability through autoregressive modeling. Extensive experiments on challenging time-dependent PDEs demonstrate that PIANO achieves state-of-the-art performance, significantly improving accuracy and stability over existing methods. We further show that PIANO outperforms existing methods in weather forecasting.

## 1 INTRODUCTION

Pierre-Simon Laplace remarked in 1814, *"We may regard the present state of a system as the effect of its past and the cause of its future"* (De Laplace, 1995). This observation reflects a core principle in classical physics: the behavior of a dynamical system is determined by its current state, with future states unfolding autoregressively. From predicting global weather to modeling heat diffusion or fluid flow, the challenge lies in understanding how the present state shapes the future. Such phenomena are mathematically described by time-dependent PDEs. Yet, modern machine learning approaches for solving PDEs such as PINNs (Raissi et al., 2019) predict states at each time step independently, without explicitly conditioning on prior states. As we show in this paper, neglecting autoregression in PINNs can lead to unstable and inaccurate predictions, underscoring the need to model dynamical systems autoregressively.

Autoregressive (AR) models are defined by a recursive structure in which the state $u(\cdot, t_n)$ at time $t_n$ is computed as a function of one or more preceding states: $u(\cdot, t_n) := f(u(\cdot, t_{n-1}), u(\cdot, t_{n-2}), \dots)$. Non-autoregressive (non-AR) models, by contrast, compute the solution independently at each time step from the input coordinates, typically as $u(\cdot, t_n) := f(\cdot, t_n)$.

PINNs solve time-dependent PDEs by training neural networks with a loss that penalizes the residuals of the governing equations. PINNs are non-AR by design and estimate the state $u(\cdot, t)$ directly from the coordinates $(\cdot, t)$ in a pointwise fashion. PINNs have been successfully applied to many physical systems, including fluid mechanics (Cai et al., 2021) and cardiovascular flows (Raissi et al., 2020), among others. Despite their success, PINNs often struggle to accurately model dynamical systems (Wang et al., 2024). To address this, recent extensions have incorporated sequential architectures to better capture temporal dependencies in the input space, as in PINNsFormer (Zhao et al., 2024) and PINNMamba (Xu et al., 2025). However, these models remain non-AR, as they are only sequential in the coordinates $(\cdot, t)$ and prediction $u(\cdot, t_n)$ at each time step $t_n$ is made independently of prior state $u(\cdot, t_{n-1})$.

In this work, we first show that PINNs suffer from temporal instability in dynamical systems, leading to error growth over time. To address this, we introduce PIANO (Physics-Informed Autoregressive Network), an autoregressive PDE solver that enforces physical laws through PDE residuals in the loss function while systematically penalizing error growth. As illustrated in Figure 1, PIANO learns

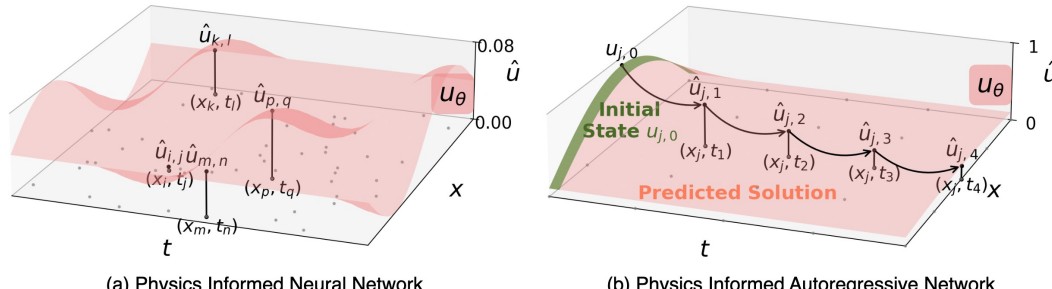

Figure 1: Comparison of (a) standard PINNs and (b) our PIANO framework on solving the Reaction equation. PINNs predict $\hat{u}(x,t)$ independently for each sample $(x,t)$, leading to trivial near-zero solutions. In contrast, PIANO conditions predictions at $t_n$ on the previous state $\hat{u}(x, t_{n-1})$, enabling stable, accurate solution propagation from the known initial state $u(x,0)$.

a state-transition function conditioned on its own past predictions, enabling accurate propagation of the solution from known initial conditions forward in time. PIANO thus provides a robust, stable, and physically consistent framework for simulating dynamical systems. Our contributions are as follows:

- **Theoretical Foundations.** We prove that PINNs are temporally unstable for time-dependent PDEs, with errors growing uncontrollably over time (Section 3).
- **A Novel Autoregressive Framework.** We propose PIANO, a physics-informed approach that reformulates PINNs to models temporal evolution autoregressively, conditioning each prediction on prior states to curb error growth (Section 4).
- **Empirical Validation.** In Section 5, we validate PIANO on challenging time-dependent PDE benchmarks and weather forecasting, demonstrating improved accuracy and stability over existing methods.

## 2 RELATED WORK

**Physics-Informed Neural Networks (PINNs)** Neural networks for solving PDEs date back several decades (Lagaris et al., 1998). The modern PINN framework (Raissi et al., 2019) uses automatic differentiation to enforce PDE residuals as soft constraints, enabling applications across fluid dynamics (Cai et al., 2021), solid mechanics (Haghighat et al., 2021), and weather forecasting (Verma et al., 2024). Standard PINNs, typically implemented as pointwise MLPs mapping $(x,t) \mapsto u(x,t)$, neglect temporal dependencies and thus suffer from compounding errors in dynamical systems (Zhao et al., 2024; Krishnapriyan et al., 2021). Neural operators (e.g., FNOs (Li et al., 2021)) are a family of data-driven surrogate models that approximate the PDE solution operator from labeled pairs, whereas PINNs solve a given PDE instance by enforcing residuals and boundary/initial conditions. Since our focus in this work is on PINNs, we benchmark primarily against PINN variants. For real-world forecasting tasks such as weather prediction, we include FNO-based methods as baselines for completeness.

**Improving PINNs for Dynamical Systems** Recent work has sought to address the limitations of PINNs on time-dependent PDEs. Wang et al. (2024) show that dynamical systems are inherently autoregressive and that standard PINNs violate this principle, leading to temporal errors; they propose a causal re-weighting of the PDE residual to restore physical causality. Other strategies include curriculum and sequence-to-sequence training (Krishnapriyan et al., 2021; Penwarden et al., 2023), as well as adaptive losses and sampling (Wight & Zhao, 2020). Li et al. (2024) introduce causality-enhanced PINNs using discretized losses and transfer learning. Architectural advances capture temporal context more explicitly, e.g., PINNsFormer (transformers) (Zhao et al., 2024), PINNMamba (state-space models) (Xu et al., 2025), and RoPINNs (spatio-temporal regions) (Wu et al., 2024). While these approaches improve temporal consistency, they remain non-autoregressive. In contrast, we propose a fully autoregressive framework that addresses this critical gap.

**Autoregressive Models and Self-Supervision** Autoregressive models, which predict the next token or state from preceding ones, underpin modern sequence modeling in NLP (Brown et al., 2020) and time-series forecasting (Das et al., 2024). Their structure is also well-suited for physical simulations (Wang et al., 2024; Li et al., 2024), but existing AR models are typically trained in a supervised setting on large datasets. Our work adapts this paradigm to the unsupervised, physics-constrained regime of PINNs. Related ideas appear in self-supervised rollouts from reinforcement learning (Hafner et al., 2019; Schrittwieser et al., 2020) and generative modeling (Ho et al., 2020), where models learn from their own predictions. PIANO is the first to bring autoregressive modeling and self-supervised rollouts into a PINN framework.

## 3 THEORETICALLY ANALYZING TEMPORAL INSTABILITY IN PINNS

This section presents preliminaries and a theoretical analysis of error propagation in PINNs for time-dependent PDEs.

### 3.1 PRELIMINARIES

We consider a differential equation defined over a domain $\Omega \subset \mathbb{R}^d$, with solution $u : \mathbb{R}^d \to \mathbb{R}^l$. The domain's interior, initial, and boundary subset are denoted by $\Omega$, $\Omega_0$, and $\partial\Omega$, respectively. Differential operators $\mathcal{O}_\Omega$, $\mathcal{O}_{\Omega_0}$, and $\mathcal{O}_{\partial\Omega}$ encode the governing equations (PDEs), initial conditions (ICs), and boundary conditions (BCs). For example, the heat equation is written as $\mathcal{O}_\Omega(u)(x) = u_t - u_{xx}$. The complete problem formulation is:

$$\mathcal{O}_\Omega(u)(x) = 0, \ x \in \Omega; \quad \mathcal{O}_{\Omega_0}(u)(x) = 0, \ x \in \Omega_0; \quad \mathcal{O}_{\partial\Omega}(u)(x) = 0, \ x \in \partial\Omega$$

PINNs (Raissi et al., 2019) approximate the solution $u$ with a neural network $u_\theta$, trained to minimize the residuals of the governing constraints:

$$\mathcal{L}(u_\theta) = \sum_{X \in \{\Omega, \Omega_0, \partial\Omega\}} \frac{\lambda_X}{N_X} \sum_{i=1}^{N_X} \left\| \mathcal{O}_X(u_\theta)(x_X^{(i)}) \right\|^2, \tag{1}$$

where $N_X$ is the number of collocation points in subset $X$, and $\lambda_X$ weights each term.

### 3.2 UNCONTROLLED ERROR PROPAGATION IN PINNS

The standard PINN formulation often fails to produce accurate solutions for time-dependent PDEs (Zhao et al., 2024; Xu et al., 2025). We argue that this is not simply an optimization issue, but a deeper architectural mismatch. Classical time-stepping schemes like finite difference or Runge–Kutta are explicitly autoregressive: they update the solution at time $t_{n+1}$ using the known state at $t_n$, preserving how dynamical systems evolve in time (Iserles, 2009; Butcher, 2016). In contrast, PINNs predict each state directly from coordinates $(\cdot, t)$ without conditioning on prior predictions, effectively breaking this autoregressive structure. Viewed through the lens of semigroup theory (Pazy, 2012), time-stepping methods approximate an evolution operator that advances the system forward, an operator that PINNs fail to represent. We now formalize this mismatch by defining the evolution operator.

**Definition 3.1** (Evolution Operator). A time-dependent PDE of the form $\frac{\partial u}{\partial t} = \mathcal{F}(u, t)$ defines a dynamical system. Its solution can be described by an evolution operator, $\mathcal{G}(\Delta t)$, which maps the state of the system at time $t_n$ to the state at time $t_{n+1} = t_n + \Delta t$:

$$u_{true}(x, t_{n+1}) = \mathcal{G}(\Delta t)[u_{true}(x, t_n)]. \tag{2}$$

A stable solver must ensure that errors do not amplify uncontrollably as this operator is applied repeatedly. Let the error of a model $u_\theta$ at time step $t_n$ be $e_n(x) = u_\theta(x, t_n) - u_{true}(x, t_n)$. We can now define the source of instability in non-AR models.

**Definition 3.2** (One-Step Rollout Error). Given a model's solution $u_\theta(x, t_n)$, the true physical evolution would yield the state $\mathcal{G}(\Delta t)[u_\theta(x, t_n)]$ at time $t_{n+1}$. The one-step rollout error is the discrepancy between the model's actual prediction at $t_{n+1}$ and the physically evolved state:

$$\delta_n = \left\| u_\theta(x, t_{n+1}) - \mathcal{G}(\Delta t)[u_\theta(x, t_n)] \right\|_2. \tag{3}$$

This error quantifies how poorly the model approximates the true one-step dynamics when initialized from its own prediction at the previous time step.

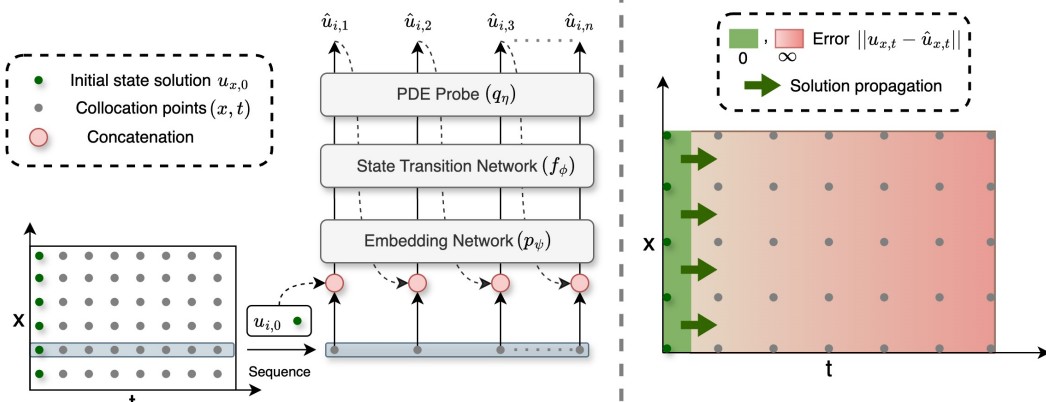

Figure 2: PIANO overview. (Left) The model processes a sequence of collocation points $(x_i, t_j)_{j=0}^n$ for each fixed $x_i$. At each time step $t_j$, the input—formed by concatenating $(x_i, t_j)$ with the previous prediction $\hat{u}(x_i, t_{j-1})$—is passed through: an embedding network $p_\psi$ to produce a high-dimensional embedding, a state transition network $f_\phi$ to model latent temporal dynamics, and a PDE probe $q_\eta$ to decode the final solution. (Right) Solution propagation from the initial state. The known initial condition $u(x, t_0)$ anchors the sequence, enabling stable and accurate learning.

**Theorem 3.3** (Error Propagation in PINNs). *For non-autoregressive PINNs, the error at step $t_{n+1}$ is bounded by the sum of the propagated error from the previous step $t_n$ and the one-step rollout error $\delta_n$:*

$$\|e_{n+1}\|_2 \leq L_\mathcal{G} \cdot \|e_n\|_2 + \delta_n, \tag{4}$$

*where $L_\mathcal{G}$ is the Lipschitz constant of the true evolution operator $\mathcal{G}(\Delta t)$.*

*Proof Sketch.* By definition, $e_{n+1} = u_\theta(x, t_{n+1}) - u_{true}(x, t_{n+1})$. Substituting $u_{true}(x, t_{n+1}) = \mathcal{G}(\Delta t)[u_{true}(x, t_n)]$ and adding and subtracting $\mathcal{G}(\Delta t)[u_\theta(x, t_n)]$, we have:

$$e_{n+1} = (u_\theta(x, t_{n+1}) - \mathcal{G}(\Delta t)[u_\theta(x, t_n)]) + (\mathcal{G}(\Delta t)[u_\theta(x, t_n)] - \mathcal{G}(\Delta t)[u_{true}(x, t_n)]).$$

Applying the triangle inequality and the Lipschitz property of $\mathcal{G}$ yields the result. The complete proof is provided in Appendix A □

*Remark* 3.4. Theorem 3.3 reveals a critical flaw in non-autoregressive PINNs. The loss function in Eq. 1 only penalizes the static PDE residual, leaving the one-step rollout error, $\delta_n$, unconstrained. This allows a new error to be introduced at each time step, which then compounds with previously propagated errors, causing long-term instability and error growth.

## 4 METHODOLOGY

We propose PIANO (Physics-Informed Autoregressive Network), a framework that addresses the temporal instability in PINNs by integrating autoregressive modeling into the PINN paradigm. This section outlines the PIANO architecture and its self-supervised training strategy.

### 4.1 ARCHITECTURE

The architecture is shown in Figure 2. The domain is defined over coordinates $(x, t) \in \Omega \subset \mathbb{R}^d$, where $x$ denotes spatial or physical variables and $t$ denotes time. The domain is discretized, and for each $x_i$, a sequence of time points $(t_j)_{j=0}^M$ is sampled. At each step $t_j$, the input is constructed by concatenating $(x_i, t_j)$ with the previous prediction $\hat{u}(x_i, t_{j-1})$. The inputs are processed by three components: an Embedding Network, a state-space Transition Network, and a PDE Probe to predict the solution $\hat{u}(x_i, t_j) \in \mathbb{R}^l$. We now describe each component in detail.

**Sampling and Input:** For each fixed $x_i$, we sample an evenly spaced temporal sequence $S_{x_i} := [(x_i, t_0), (x_i, t_1), \ldots, (x_i, t_M)]$ from the domain $\Omega$, where $t_0 \in \Omega_0$ is the initial time and $t_M$ the final step. At each $(x_i, t_j) \in S_{x_i}$, the input vector $s_j^i := (x_i, t_j, \hat{u}(x_i, t_{j-1})) \in \mathbb{R}^{d+l}$ is constructed by concatenating the coordinate with the previous prediction.

**Embedding Network ($p_\psi$):** An embedding network $p_\psi$ maps each input vector $s_j^i$ to a higher-dimensional representation $m_j^i \in \mathbb{R}^k$, enhancing expressiveness and preparing the sequence for further processing.

**State Transition Network ($f_\phi$):** The transition network $f_\phi$ models latent dynamics by maintaining a hidden state $h$ that summarizes past information. It follows a discrete-time state-space formulation with two steps:

$$h_j^i = \sigma(\text{LN}(\mathbf{A}h_{j-1}^i + \mathbf{B}m_j^i))$$
$$o_j^i = \mathbf{C}h_j^i + \mathbf{D}m_j^i + m_j^i,$$

where $h_j^i \in \mathbb{R}^k$ is the recurrent hidden state and $o_j^i \in \mathbb{R}^k$ is the output representation. The function $f_\phi$ is parameterized by learnable matrices $\mathbf{A}, \mathbf{B}, \mathbf{C}, \mathbf{D} \in \mathbb{R}^{k \times k}$, along with a nonlinearity $\sigma$ and layer normalization (LN).

**PDE Probe ($q_\eta$):** Finally, the probe decodes each output:

$$\hat{u}(x_i, t_j) = q_\eta(o_j^i), \quad o_j^i \in \mathbb{R}^k \mapsto \hat{u}(x_i, t_j) \in \mathbb{R}^l.$$

$q_\eta$ is applied independently at each time step, translating latent features into the PDE solutions.

## 4.2 PHYSICS-INFORMED EXPERIENCE LEARNING

We propose Physics-Informed Experience Learning (PIEL), a training paradigm where the model improves by enforcing physical consistency over its own rollouts. Starting from the known initial condition $u(x, t_0)$, PIANO autoregressively predicts the trajectory by conditioning each new state on the previous one, with gradients propagated through time via backpropagation through time (BPTT).

For each spatial location $x_i$, the rollout is:

$$\hat{u}(x_i, t_j) = u_\theta(x_i, t_j, \hat{u}(x_i, t_{j-1})), \quad j = 1, \ldots, M,$$

where $\theta = \{\psi, \phi, \eta\}$ denotes model parameters.

The computational graph is discrete in time and not connected across neighboring spatial points. Hence, we approximate all PDE derivatives using second-order accurate finite differences applied over the full predicted solution grid.

For region $X \in \{\Omega, \partial\Omega\}$, we define the residual energy:

$$\mathcal{E}_X(x_i, u_\theta) = \frac{1}{M} \sum_{j=1}^M \|\mathcal{O}_X[\hat{u}](x_i, t_j)\|^2, \tag{5}$$

where $\mathcal{O}_X[\hat{u}]$ denotes the PDE residual (via second-order finite differences) for $X = \Omega$ and the boundary condition error for $X = \partial\Omega$.

The total training loss aggregates these contributions:

$$\mathcal{L}_{\text{PIANO}} = \sum_{X \in \{\Omega, \partial\Omega\}} \frac{\lambda_X}{N_X} \sum_{i=1}^{N_X} \mathcal{E}_X(x_i, u_\theta), \tag{6}$$

where $\lambda_X$ weights PDE and boundary contributions. When data is available, teacher forcing with a data loss can be added, allowing PIEL to operate in both white-box (physics-only) and grey-box (physics+data) settings. Appendix B provides the full PIEL training algorithm for completeness. We now provide theoretical bound on one-step rollout error in PIANO.

Table 1: PIANO as a robust and accurate PDE solver across a range of PDE benchmarks. rMAE and rRMSE are reported separately for each PDE. Best values are highlighted in **bold** and the second best are underlined. PIANO outperforms baselines across all benchmarks. Promotion refers to the relative error reduced w.r.t. the second best model $\left(1 - \frac{\text{Our Error}}{\text{Second Best Error}}\right)$.

| Model | Wave | | Reaction | | Convection | | Heat | |
|---|---|---|---|---|---|---|---|---|
| | rMAE | rRMSE | rMAE | rRMSE | rMAE | rRMSE | rMAE | rRMSE |
| PINNs (JCP'19) | 0.4101 | 0.4141 | 0.9803 | 0.9785 | 0.8514 | 0.8989 | 0.8956 | 0.9404 |
| QRes (ICDM'21) | 0.5349 | 0.5265 | 0.9826 | 0.9830 | 0.9035 | 0.9245 | 0.8381 | 0.8800 |
| FLS (TAI'22) | 0.1020 | 0.1190 | 0.0220 | 0.0390 | 0.1730 | 0.1970 | 0.7491 | 0.7866 |
| PINNsFormer (ICLR'24) | 0.3559 | 0.3632 | 0.0146 | 0.0296 | 0.4527 | 0.5217 | 0.2129 | 0.2236 |
| RoPINNs (NeurIPS'24) | 0.1650 | 0.1720 | 0.0070 | 0.0170 | 0.6350 | 0.7200 | 0.1545 | 0.1622 |
| KAN (ICLR'25) | 0.1433 | 0.1458 | 0.0166 | 0.0343 | 0.6049 | 0.6587 | 0.0901 | 0.1042 |
| PINNMamba (ICML'25) | 0.0197 | 0.0199 | 0.0094 | 0.0217 | 0.0188 | 0.0201 | 0.0535 | 0.0583 |
| **PIANO (ours)** | **0.0057** | **0.0059** | **0.0001** | **0.0008** | **0.0032** | **0.0104** | **0.0000** | **0.0002** |
| Promotion (%) | 71.1 | 70.4 | 98.6 | 95.3 | 83.0 | 48.3 | 100.0 | 99.7 |

**Theorem 4.1** (Bound on One-Step Rollout Error in PIANO). *Let $\mathcal{G}(\Delta t)$ denote the exact evolution operator of the PDE, and let $\mathcal{G}_{h,\Delta t}$ be a finite-difference approximation of temporal order $p$ and spatial order $q$ with discretization constant $\kappa > 0$. Here $\Delta t > 0$ is the temporal step size and $h > 0$ is the spatial grid spacing. If PIANO is trained with the PIEL loss in Eq. 6, and $\rho$ denotes the residual energy controlled by $\mathcal{L}_{PIANO}$, then the one-step rollout error (Definition 3.2) satisfies*

$$\delta_n \leq \rho + \kappa\left(\Delta t^p + h^q\right), \quad \text{for all } n = 0, \ldots, M-1. \tag{7}$$

*Sketch Proof.* We decompose

$$\delta_n = \|\hat{u}(x, t_{n+1}) - \mathcal{G}(\Delta t)[\hat{u}(x, t_n)]\|_2 \leq \|\hat{u}(x, t_{n+1}) - \mathcal{G}_{h,\Delta t}(\hat{u}(x, t_n))\|_2$$
$$+ \|\mathcal{G}_{h,\Delta t}(\hat{u}(x, t_n)) - \mathcal{G}(\Delta t)[\hat{u}(x, t_n)]\|_2.$$

The first term is directly penalized by the PIEL objective and bounded by $\rho$. The second term is the consistency error of the $(p, q)$ finite-difference scheme, bounded by $\kappa(\Delta t^p + h^q)$. $\square$

*Remark* 4.2. In PIANO the initial error is zero ($e_0 = 0$), since rollouts are propagated from the true initial condition. For the second-order discretization used in practice ($p = q = 2$), the consistency term $\kappa(\Delta t^2 + h^2)$ vanishes under mesh refinement. Thus, the dominant contribution to $\delta_n$ is the residual $\rho$, which is minimized during training via Eq. 6. Consequently, $\delta_n$ can be made arbitrarily small by improving training (reducing $\rho$) and refining the discretization ($\Delta t, h \to 0$).

## 5 EXPERIMENTS

To empirically demonstrate the effectiveness of PIANO, we evaluate its performance on PDE benchmarks (Section 5.1) and a real-world weather forecasting task (Section 5.2).

### 5.1 PDE BENCHMARKS

**Benchmarks** We evaluate PIANO on four time-dependent PDE benchmarks: the Wave, Reaction, Convection, and Heat equations. These benchmarks are widely used in the literature (Zhao et al., 2024; Xu et al., 2025) and span diverse numerical challenges: higher-order derivatives (Wave), nonlinear dynamics (Reaction), numerical stiffness (Heat), and transport-dominated behavior prone to numerical diffusion (Convection). Detailed description of each PDE is provided in Appendix C.1.

**Baselines** We benchmark PIANO against a broad set of baselines: classical PINN variants (MLP-based PINNs (Raissi et al., 2019), First-Layer Sine networks (FLS) (Wong et al., 2022), and Quadratic Residual Networks (QRes) (Bu & Karpatne, 2021)); recent advances (Kolmogorov–Arnold Networks (KANs) (Liu et al., 2025) and Region-Optimized PINNs (RoPINNs)

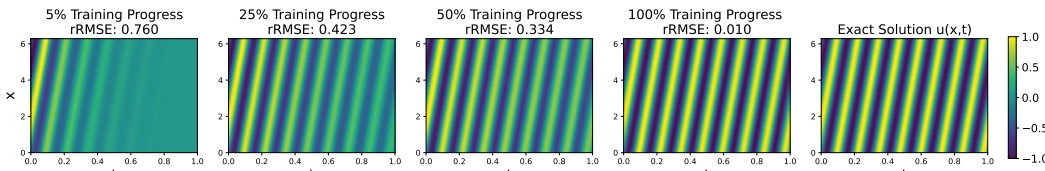

Figure 3: Training dynamics of PIANO on the convection equation. At 5%, predictions are accurate only near the initial state; by 25–50% temporal propagation improves, and at convergence prediction is visually indistinguishable from the exact solution. This confirms that propagating correct solutions from known initial conditions ensures stable convergence without intermediate failure.

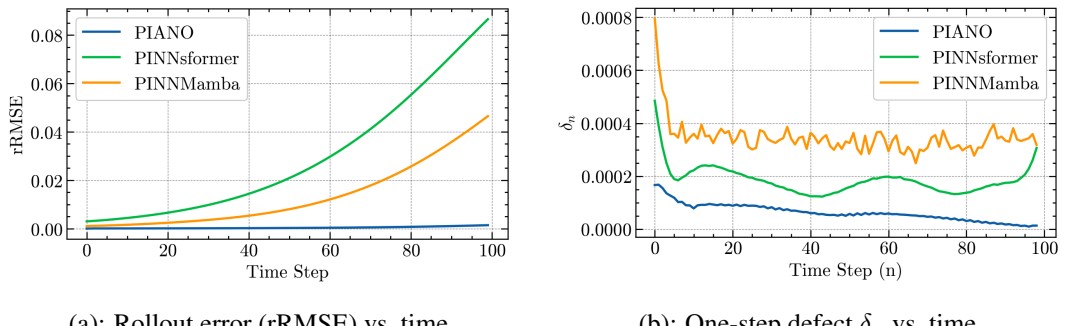

(a): Rollout error (rRMSE) vs. time.          (b): One-step defect $\delta_n$ vs. time.

Figure 4: Temporal stability diagnostics. PIANO maintains near-flat error growth and the smallest one-step defect, while PINNsFormer and PINNMamba exhibit rapidly increasing rollout error and larger $\delta_n$. Lower is better in both panels.

(Wu et al., 2024)); and state-of-the-art sequential models (PINNsFormer (Zhao et al., 2024) and PINNMamba (Xu et al., 2025)), which are sequential but not autoregressive—ideal for testing PIANO's autoregressive advantage. Full baseline descriptions are provided in Appendix C.2.

**Implementation Details**   PIANO is implemented as a state-space architecture (Section 4) and trained on a $200 \times 200$ discretized spatio-temporal grid using the AdamW optimizer. All models are trained on approx. same number of samples. For fairness, baselines rely on official implementations with reported hyperparameters and training routines. Regarding computational cost, PIANO is comparable to sequential baselines like PINNsFormer and PINNMamba, while its simpler state-space architecture offers efficiency gains (Appendix C.3). Performance is measured using relative Mean Absolute Error (rMAE) and relative Root Mean Squared Error (rRMSE), which are standard metrics in the PINN literature (Xu et al., 2025). Full detail on hyperparameters and implementation are provided in Appendix C.4.

**Results**   Table 1 summarizes the PDE benchmark results. Pointwise PINNs suffer from high errors (consistent with Theorem 3.3), while sequential models such as PINNMamba perform better. PIANO, however, consistently sets a new state of the art across all four benchmarks. For the Reaction and Heat equations, errors are driven to near zero with about 100% promotion over the second best model, while on the more challenging Wave and Convection equations PIANO outperforms PINNMamba by 70–80% promotion. These results highlight the accuracy and robustness of our autoregressive formulation for time-dependent PDEs. A two-tailed $t$-test ($p < 0.05$) confirms PIANO's improvements are statistically significant (Appendix C.5).

**Training Dynamics**   Figure 3 shows PIANO learning the convection equation during training. Starting from the initial state, predictions are initially localized (5%), gradually extend forward but remain blurry (25–50%), and at convergence (100%) become visually indistinguishable from the ground truth (rRMSE 0.010). This confirms the intuition that propagating the correct solution from known initial conditions enables stable convergence without failure in between. Additional examples on other PDEs are provided in Appendix C.6.

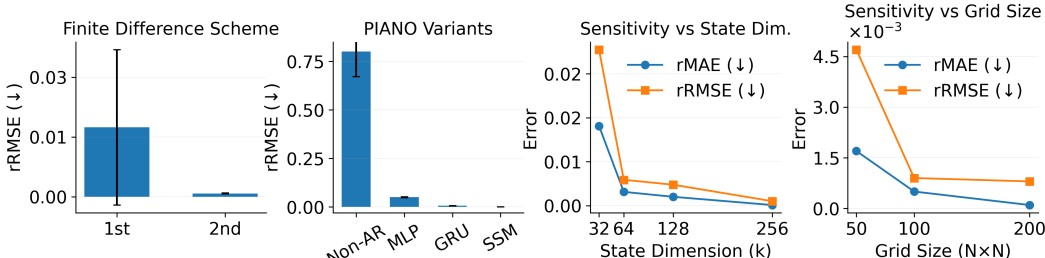

Figure 5: Ablations and hyperparameter guidance. The plots show the effect of finite difference order, the performance of different PIANO variants, and sensitivity to state dimension and training grid resolution. Error bars denote standard deviations across ten runs.

**Temporal Stability Diagnostics** We perform stability diagnostics to validate the theoretical arguments using the Reaction. Results are reported in Figure 4. Panel (a) shows rollout error (rRMSE) over time. Baselines suffer from uncontrolled error propagation, with errors compounding and growing rapidly as time advances. In contrast, PIANO maintains nearly flat error growth throughout the rollout. Panel (b) reports the one-step defect $\delta_n$ defined in Def. 3.2. PIANO consistently achieves the smallest $\delta_n$ by explicitly penalizing this quantity during training, while the baselines accumulate larger defects over time. These results validate our theoretical analysis: non-AR PINNs are inherently unstable with exponential error growth, whereas PIANO's autoregressive design ensures stability and accurate long-term predictions.

**Ablations and Hyperparameter Guidance.** Figure 5 summarizes the ablation and hyperparameter guidance for PIANO. The experiments are conducted on the Reaction equation. The ablation studies demonstrate that a second-order finite difference scheme yields substantially lower error than the first-order version, confirming the importance of accurate derivative approximations. They also show that autoregression is critical: while a non-autoregressive baseline performs poorly, progressively richer backbones (MLP, GRU, SSM) with PIANO deliver significant gains, with the SSM achieving the lowest error. The sensitivity analysis further highlights that increasing the state dimension and training grid resolution consistently reduces error, with diminishing returns once $k = 256$ and a $200 \times 200$ grid are reached. Together, these findings validate the design of PIANO and provide practical guidance for hyperparameter choices in PDE learning. Extended analysis is presented in Appendix C.7.

## 5.2 GLOBAL WEATHER FORECASTING

**Background.** Global weather forecasting is the task of predicting the future evolution of key atmospheric variables (e.g., temperature, winds, pressure) across the entire Earth. Numerical simulations of atmospheric physics remain the standard for weather forecasting but are computationally demanding. Deep learning has emerged as a faster alternative, yet most models operate as black boxes that ignore physical laws, leading to unstable or unphysical predictions. Physics-informed approaches aim to bridge this gap by embedding governing equations into neural networks. ClimODE (Verma et al., 2024), for example, formulates weather evolution as a neural ODE constrained by the advection PDE, which models how quantities such as temperature and pressure are transported by winds. This inductive bias yields forecasts that are both stable and physically consistent. Building on this idea, PIANO introduces an autoregressive training scheme to further improve long-term accuracy and stability.

**Setup.** The forecasting task involves predicting the temporal evolution of five key atmospheric variables from the ERA5 (Rasp et al., 2020) dataset: atmospheric temperature (`t`), surface temperature (`t2m`), horizontal wind components (`u10`, `v10`), and geopotential height (`z`). Following ClimODE, we adopt a physics-informed framework with two coupled components: (i) an advection term that enforces conservation of transported quantities, and (ii) a neural network $f_\theta$ that learns to update the velocity field from the current state and its spatial gradients. PIANO retains the physics-informed structure of ClimODE but trains autoregressively with teacher forcing. Instead of predict-

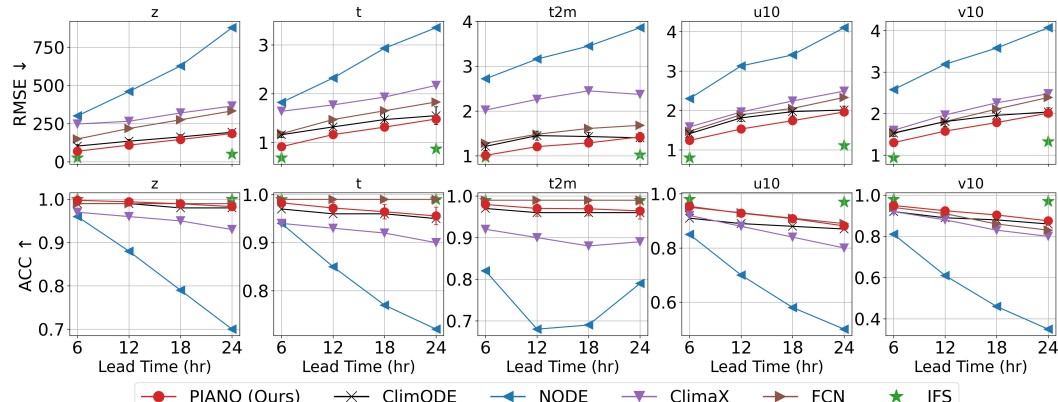

Figure 6: Comparison of global weather forecasting performance on the ERA5 dataset. The plot shows the latitude-weighted root mean square error (RMSE) and anomaly correlation coefficient (ACC) for PIANO against recent baselines over lead times from 6 to 24 hours. The task involves predicting five key atmospheric variables: geopotential (z), atmospheric temperature (t), 2-meter surface temperature (t2m), and the 10-meter U-wind (u10) and V-wind (v10) components. PIANO consistently achieves lower error across all variables, showcasing the effectiveness of its autoregressive physics-informed framework for modeling complex dynamical systems.

ing an entire trajectory in one step, each forecast is conditioned on the true state at the previous time point. At inference, we use free-rollout forecasting: the model advances autoregressively using its own predictions at each step, without teacher forcing.

We evaluate on ERA5, which provides 6-hourly reanalysis data at 5.625° resolution for the five variables listed above. Comparisons are made against NODE (Verma et al., 2024), FCN (Pathak et al., 2022), ClimaX (Nguyen et al., 2023), and ClimODE (Verma et al., 2024), with the IFS (ECMWF, 2023) serving as the gold-standard numerical baseline. Performance is assessed using two latitude-weighted metrics: Root Mean Square Error (RMSE), which measures absolute prediction error, and Anomaly Correlation Coefficient (ACC), which quantifies directional accuracy by correlating predicted and observed anomalies. Extended setup and implementation details are provided in Appendix D.1.

**Results** Figure 6 compares PIANO against baselines on RMSE and ACC across five key atmospheric variables over 6 to 24-hour lead times. PIANO consistently achieves a lower RMSE across all variables and horizons, indicating improved forecast accuracy. Notably, its performance gains are most significant at shorter lead times, where the autoregressive use of observed initial states effectively limits error propagation. These results highlight superior performance of PIANO for ERA5 forecasting, highlighting the benefits of combining an autoregressive training strategy with a physics-informed framework for simulating complex dynamical systems. The complete result table is provided in Appendix D.2.

## 6 CONCLUSION

We present PIANO, a physics-informed autoregressive framework for solving time-dependent PDEs. Our theoretical analysis demonstrates that non-autoregressive PINN formulations are unstable and accumulate errors. By aligning model design with the autoregressive property of dynamical systems, PIANO mitigates the error accumulation seen in conventional PINNs and provides a stable foundation for learning physical dynamics. Experiments on a challenging PDE benchmark show that PIANO achieves state-of-the-art accuracy. It also improves physics-informed methods for global weather forecasting. Beyond these gains, PIANO points to a broader direction: physics-informed learning can benefit significantly from architectures that respect the temporal evolution of the systems they model. Extending this approach to multi-scale processes and real-world scientific applications presents an exciting avenue for future research.

## REPRODUCIBILITY STATEMENT

All PDE benchmarks used in this work are described in detail in Appendix C.1, including governing equations, domains, and analytical solutions. Implementation details, training procedures, hyperparameters, and complexity analysis are provided in Appendix C.4 and Appendix C.3, with full training algorithms given in Appendix B. Extended proofs and theoretical results are documented in Appendix A. For empirical evaluation, we report both single-run and multi-run statistics, with significance testing in Appendix C.5. Additional qualitative results and training dynamics are included in Appendix C.6, along with guidance on hyperparameters and ablations in Appendix C.7. The ERA5 dataset used for global weather forecasting is publicly available, and our preprocessing steps and forecasting setup are described in Appendix D.1. An anonymous code repository containing implementations of PIANO and scripts to reproducing experiments is available at `https://anonymous.4open.science/r/piano_iclr-73C8`.

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

# A   PROOFS AND EXTENDED THEORETICAL ANALYSIS

## A.1   PROOFS

**Theorem A.1** (Error Propagation in PINNs). *For non-autoregressive PINNs, the error at step $t_{n+1}$ is bounded by the sum of the propagated error from the previous step $t_n$ and the one-step rollout error $\delta_n$:*

$$\|e_{n+1}\|_2 \leq L_{\mathcal{G}} \cdot \|e_n\|_2 + \delta_n, \tag{8}$$

*where $L_{\mathcal{G}}$ is the Lipschitz constant of the true evolution operator $\mathcal{G}(\Delta t)$.*

*Proof.* The proof relies on the key assumption that the true evolution operator, $\mathcal{G}(\Delta t)$, is Lipschitz continuous with a constant $L_{\mathcal{G}}$. This property ensures that the operator does not excessively amplify differences between input states, and it is formally stated as:

$$\|\mathcal{G}(\Delta t)[a] - \mathcal{G}(\Delta t)[b]\|_2 \leq L_{\mathcal{G}} \cdot \|a - b\|_2 \tag{9}$$

for any two system states $a$ and $b$.

We begin with the definition of the error at time step $t_{n+1}$:

$$e_{n+1} = u_\theta(x, t_{n+1}) - u_{true}(x, t_{n+1}). \tag{10}$$

By definition, the true solution at $t_{n+1}$ is given by the evolution operator $\mathcal{G}(\Delta t)$ applied to the true solution at $t_n$. Substituting this into our error expression gives:

$$e_{n+1} = u_\theta(x, t_{n+1}) - \mathcal{G}(\Delta t)[u_{true}(x, t_n)]. \tag{11}$$

We now add and subtract the term $\mathcal{G}(\Delta t)[u_\theta(x, t_n)]$. This allows us to connect the model's prediction at $t_{n+1}$ to the evolution of its own prediction from $t_n$:

$$e_{n+1} = (u_\theta(x, t_{n+1}) - \mathcal{G}(\Delta t)[u_\theta(x, t_n)]) \\ + (\mathcal{G}(\Delta t)[u_\theta(x, t_n)] - \mathcal{G}(\Delta t)[u_{true}(x, t_n)]). \tag{12}$$

Taking the $L_2$ norm and applying the triangle inequality ($\|A + B\| \leq \|A\| + \|B\|$) yields:

$$\|e_{n+1}\|_2 \leq \|u_\theta(x, t_{n+1}) - \mathcal{G}(\Delta t)[u_\theta(x, t_n)]\|_2 \\ + \|\mathcal{G}(\Delta t)[u_\theta(x, t_n)] - \mathcal{G}(\Delta t)[u_{true}(x, t_n)]\|_2. \tag{13}$$

We recognize the first term on the right-hand side as the definition of the one-step rollout error, $\delta_n$. For the second term, we apply the Lipschitz continuity of the operator $\mathcal{G}$:

$$\|\mathcal{G}(\Delta t)[u_\theta(x, t_n)] - \mathcal{G}(\Delta t)[u_{true}(x, t_n)]\|_2 \\ \leq L_{\mathcal{G}} \cdot \|u_\theta(x, t_n) - u_{true}(x, t_n)\|_2 \\ = L_{\mathcal{G}} \cdot \|e_n\|_2. \tag{14}$$

Substituting these two results back into Equation equation 13, we arrive at the final inequality:

$$\|e_{n+1}\|_2 \leq \delta_n + L_{\mathcal{G}} \cdot \|e_n\|_2. \tag{15}$$

Rearranging the terms gives the statement of the theorem:

$$\|e_{n+1}\|_2 \leq L_{\mathcal{G}} \cdot \|e_n\|_2 + \delta_n. \tag{16}$$

$\square$

**Theorem A.2** (Bound on One-Step Rollout Error in PIANO). *Let $G(\Delta t)$ be the exact evolution operator and let $G_{h,\Delta t}$ be a finite-difference (FD) approximation of temporal order $p$ and spatial order $q$, with consistency constant $\kappa > 0$. Let $L_{\text{PIANO}}$ be the PIEL loss in Eq. (6) and define*

$$\rho := C_{\text{disc}}(\Delta t, h) \, L_{\text{PIANO}}^{1/2},$$

*where $C_{\text{disc}}(\Delta t, h)$ collects the FD stencil constants and converts residual units to the state norm used below. Then for the one-step rollout error $\delta_n := \|\hat{u}(\cdot, t_{n+1}) - G(\Delta t)[\hat{u}(\cdot, t_n)]\|$ (Def. 3.2), we have for all $n = 0, \ldots, M-1$,*

$$\delta_n \leq \rho + \kappa\big(\Delta t^p + h^q\big).$$

*Proof.* Fix a step $n$ and abbreviate $\hat{u}^n := \hat{u}(\cdot, t_n)$. By adding and subtracting $G_{h,\Delta t}(\hat{u}^n)$,

$$\delta_n = \big\|\hat{u}^{n+1} - G(\Delta t)[\hat{u}^n]\big\| \leq \underbrace{\big\|\hat{u}^{n+1} - G_{h,\Delta t}(\hat{u}^n)\big\|}_{\text{(I)}} + \underbrace{\big\|G_{h,\Delta t}(\hat{u}^n) - G(\Delta t)[\hat{u}^n]\big\|}_{\text{(II)}}. \quad (17)$$

Let $\mathcal{R}_{h,\Delta t}[\hat{u}](x_i, t_j)$ denote the FD residual of the governing PDE evaluated on the predicted rollout (Eq. (5)), i.e.,

$$\mathcal{R}_{h,\Delta t}[\hat{u}](x_i, t_j) = D_t \hat{u}(x_i, t_j) - \mathcal{F}_h\big(\hat{u}(\cdot, t_j)\big)\Big|_{x_i},$$

where $D_t$ is the chosen temporal difference operator and $\mathcal{F}_h$ the spatial FD discretization of $\mathcal{F}$. For a one-step scheme, the FD update can be written as

$$G_{h,\Delta t}(\hat{u}^n) = \hat{u}^n + \Delta t \, \mathcal{A}_{h,\Delta t}(\hat{u}^n),$$

for some (possibly nonlinear) discrete operator $\mathcal{A}_{h,\Delta t}$ induced by the stencil. A standard algebraic manipulation of the stencil (equivalently, a discrete variation-of-constants identity) yields a bounded linear map $\mathcal{B}_{h,\Delta t}$ such that

$$\hat{u}^{n+1} - G_{h,\Delta t}(\hat{u}^n) = \mathcal{B}_{h,\Delta t}(\mathcal{R}_{h,\Delta t}[\hat{u}](\cdot, t_{n+\theta})), \quad (18)$$

for some $\theta \in \{0, 1/2, 1\}$ depending on the time stencil (forward/Crank–Nicolson/backward). On the discrete grid, all norms are equivalent and $\|\mathcal{B}_{h,\Delta t}\| \leq C_{\text{disc}}(\Delta t, h)$ for a constant that depends only on the stencil and $(\Delta t, h)$. Taking norms in Eq.18 and averaging over spatial points $\{x_i\}_{i=1}^{N_\Omega}$ gives

$$\text{(I)} \leq C_{\text{disc}}(\Delta t, h) \left(\frac{1}{N_\Omega} \sum_{i=1}^{N_\Omega} \big\|\mathcal{R}_{h,\Delta t}[\hat{u}](x_i, t_{n+\theta})\big\|^2\right)^{1/2}.$$

By definition of the residual energies and the training objective (Eq. (5)–(6)), the bracketed quantity is controlled by $L_{\text{PIANO}}$ up to the weights $\lambda_X/N_X$ and the boundary terms. Thus,

$$\text{(I)} \leq C_{\text{disc}}(\Delta t, h) \, L_{\text{PIANO}}^{1/2} = \rho.$$

*Units/Scaling.* Since $\mathcal{R}_{h,\Delta t}$ has the units of the PDE operator, $C_{\text{disc}}$ carries the complementary units to yield the state norm; therefore $\rho$ has the units of $u$. Because $L_{\text{PIANO}}$ aggregates $\frac{\lambda_X}{N_X} \sum_i E_X(x_i, \hat{u})$, the dependence on $\lambda_X, N_X$ enters only as $L_{\text{PIANO}}^{1/2} \propto \sqrt{\lambda_X/N_X}$.

By the $(p, q)$-order consistency of $G_{h,\Delta t}$ with $G(\Delta t)$ applied to the same input state $\hat{u}^n$, there exists $\kappa > 0$ such that

$$\text{(II)} \leq \kappa\big(\Delta t^p + h^q\big).$$

Substituting the bounds for (I) and (II) into Eq.17 gives

$$\delta_n \leq \rho + \kappa\big(\Delta t^p + h^q\big),$$

as claimed. $\qquad\square$

## A.2 Existence, Uniqueness and Regularity of Solutions for Evolution Operators

We study time-dependent PDEs of the form

$$\frac{du}{dt} = -Au + f(u), \quad u(0) = u_0 \in H, \tag{19}$$

where $H$ is a complete normed vector space with an inner product (Hilbert space), $-A : D(A) \subset H \to H$ is a linear operator generating a strongly continuous semigroup, and $f : H \to H$ is a nonlinear operator.

### A.2.1 Semigroup Setup and Mild Solutions

**Definition A.3** (Semigroup Generator). Let $-A : D(A) \subset H \to H$ be a closed, densely defined linear operator such that $-A$ generates a strongly continuous solution operator $\{S(t)\}_{t \geq 0} \subset \mathcal{L}(H)$; and $\{S(t)\}_{t \geq 0}$ satisfies :

1. $S(t)u_0(\cdot) = u(\cdot, t)$

2. $S(t)$ is a semigroup which satisifies $S(0) = I$ (the identity) and $S(t + s) = S(t)S(s)$ for all $t, s \geq 0$;

3. $\|S(t)\| \leq 1$ for all $t \geq 0$.

Starting at time $t = 0$, we have $u(\cdot, 0) = S(0)u_0(\cdot)$. You obtain $u(\cdot, s + t)$ by first flowing forward in time by $s$ and then flow forward by time $t$ using $u(\cdot, s)$ as initial data.

**Definition A.4** (Mild Solution). A function $u \in C([0, T], H)$ is a *mild solution* to equation 19 if it satisfies the variation of constants formula:

$$u(\cdot, t) = S(t)u_0(\cdot) + \int_0^t S(t - s)f(u(\cdot, s)) \, ds. \tag{20}$$

**Theorem A.5** (Existence and Uniqueness of Mild Solutions). *Suppose $f : H \to H$ is Lipschitz and satisfies:*

$$\|f(u) - f(v)\| \leq L\|u - v\|, \tag{21}$$

$$\|f(u)\| \leq L(1 + \|u\|), \quad \forall u, v \in H. \tag{22}$$

*Then for any $u_0 \in H$, there exists a unique mild solution $u \in C([0, T], H)$, and:*

$$\|u(t)\| \leq C_T(1 + \|u_0\|), \quad \forall t \in [0, T].$$

*Proof.* Let $X := C([0, T], H)$ with norm $\|u\|_X := \sup_{t \in [0,T]} \|u(t)\|$. Define the mapping $\mathcal{J} : X \to X$ by

$$(\mathcal{J}u)(t) := S(t)u_0 + \int_0^t S(t - s)f(u(s)) \, ds.$$

We show $\mathcal{J}$ is a contraction for small $T$:

$$\|\mathcal{J}u - \mathcal{J}v\|_X \leq \sup_{t \in [0,T]} \int_0^t \|f(u(s)) - f(v(s))\| ds$$

$$\leq LT\|u - v\|_X.$$

Choose $T < 1/L$ so that $\mathcal{J}$ is a contraction. By Banach's fixed-point theorem, there exists a unique fixed point $u \in X$. Repeating over intervals gives global existence.

For the bound, using $\|S(t)\| \leq 1$:

$$\|u(t)\| \leq \|u_0\| + \int_0^t L(1 + \|u(s)\|) ds.$$

Apply Grönwall's inequality to conclude. □

### A.2.2 Regularity and Temporal Smoothness

**Theorem A.6** (Temporal Regularity). *Let $u_0 \in D(A^\gamma)$ for some $\gamma \in (0, 1]$, and let $u(t)$ be the mild solution. Then for any $0 \leq t_1 \leq t_2 \leq T$ and $\epsilon > 0$, there exists $C > 0$ such that:*

$$\|u(t_2) - u(t_1)\| \leq C|t_2 - t_1|^\theta (1 + \|u_0\|_{D(A^\gamma)}),$$

*where $\theta = \min\{\gamma, 1 - \epsilon\}$.*

*Sketch.* Write $u(t_2) - u(t_1) = I + II$, with:

$$I := S(t_2)u_0 - S(t_1)u_0,$$

$$II := \int_0^{t_2} S(t_2 - s)f(u(s))ds - \int_0^{t_1} S(t_1 - s)f(u(s))ds.$$

Estimate $I$ via semigroup regularity:

$$\|I\| \leq C|t_2 - t_1|^\gamma \|u_0\|_{D(A^\gamma)}.$$

For $II$, split as:

$$II_1 := \int_0^{t_1} [S(t_2 - s) - S(t_1 - s)]f(u(s))ds,$$

$$II_2 := \int_{t_1}^{t_2} S(t_2 - s)f(u(s))ds,$$

and use continuity of $S(t)$ and boundedness of $f(u(s))$ to obtain:

$$\|II\| \leq C|t_2 - t_1|^\theta (1 + \|u_0\|).$$

$\square$

### A.2.3 Evolution Operator Approximation

**Definition A.7** (Evolution Operator $\mathcal{G}$). Let $\Delta t > 0$. Define a learned operator $\mathcal{G} : H \to H$ such that:

$$\mathcal{G}(u_n) \approx u_{n+1}, \quad \text{where } u_n \approx u(n\Delta t), \ u_{n+1} \approx u((n + 1)\Delta t).$$

Let $\Phi_{\Delta t}(u)$ denote the exact flow:

$$\Phi_{\Delta t}(u) := S(\Delta t)u + \int_0^{\Delta t} S(\Delta t - s)f(u(s))ds.$$

**Theorem A.8** (Error Propagation of $\mathcal{G}$). *Let $u(\cdot, t)$ be the mild solution to equation 19 with $u_0 \in D(A^\gamma)$ for some $\gamma \in (0, 1]$. Suppose:*

$$\|\mathcal{G}(u) - \Phi_{\Delta t}(u)\| \leq \varepsilon(\Delta t), \quad \forall u \in \mathcal{B} \subset H.$$

*Define $\tilde{u}_0 = u_0$, and recursively $\tilde{u}_{n+1} = \mathcal{G}(\tilde{u}_n)$. Then for $u_n := u(n\Delta t)$,*

$$\|\tilde{u}_n - u_n\| \leq C_T \left( \varepsilon(\Delta t) + \Delta t^\theta (1 + \|u_0\|_{D(A^\gamma)}) \right),$$

*where $\theta = \min\{\gamma, 1 - \epsilon\}$ and $C_T$ depends on $T$ and $L$.*

*Proof.* We proceed by induction.

Base case: $\tilde{u}_0 = u_0 \Rightarrow \|\tilde{u}_0 - u_0\| = 0$.

Inductive step:

$$\begin{aligned}
\|\tilde{u}_{n+1} - u_{n+1}\| &\leq \|\mathcal{G}(\tilde{u}_n) - \Phi_{\Delta t}(\tilde{u}_n)\| \\
&\quad + \|\Phi_{\Delta t}(\tilde{u}_n) - \Phi_{\Delta t}(u_n)\| \\
&\leq \varepsilon(\Delta t) + L_\Phi \|\tilde{u}_n - u_n\|.
\end{aligned}$$

Apply recursively and use the regularity bound from A.6 for $\|u_{n+1} - \Phi_{\Delta t}(u_n)\|$ to close the estimate.

$\square$

While PIANO mitigates the recurrence error by minimizing the one-step rollout term, Theorem A.8 proves that autoregressive models that approximate the true evolution operator enjoy a provably bounded global error.

## B  TRAINING ALGORITHM

---

**Algorithm 1** Training PIANO via Experience Learning

---

1:  Initialize model parameters $\psi, \phi, \eta$.
2:  **for** each training iteration **do**
3:      Sample a batch of spatial coordinates $\{x_i\}_{i=1}^{N_x} \subset \Omega \cup \partial\Omega$.
4:      Set initial state from the known condition: $\hat{u}(x_i, t_0) \leftarrow u(x_i, t_0)$ for all $i$.
5:      Initialize hidden state $h_0^i \leftarrow \mathbf{0}$ for all $i$.
6:      **for** each time step $t_j$, for $j = 1, \ldots, M$ **do**
7:          Form input vector: $s_j^i = (x_i, t_j, \hat{u}(x_i, t_{j-1}))$.
8:          Compute embedding: $m_j^i = p_\psi(s_j^i)$.
9:          Update hidden state and output representation: $(h_j^i, o_j^i) = f_\phi(h_{j-1}^i, m_j^i)$.
10:         Predict solution: $\hat{u}(x_i, t_j) = q_\eta(o_j^i)$.
11:         Compute loss using Eq. 6
12:     **end for**
13:     Normalize loss over time steps and batch size.
14:     Update parameters $\psi, \phi, \eta$.
15: **end for**

---

This training procedure, which we refer to as Physics-Informed Experience Learning (PIEL), optimizes the model to generate physically consistent solution trajectories based on its own predictions. The experience learning component comes from the autoregressive rollout described in Algorithm 1: for each spatial coordinate $x_i$ in a batch, the model generates an entire temporal trajectory starting from the known initial condition $u(x_i, t_0)$. Each subsequent prediction $\hat{u}(x_i, t_j)$ is conditioned on the model's own previous output $\hat{u}(x_i, t_{j-1})$, which forces the model to learn from its own generated experience.

The physics-informed component governs the optimization process. Instead of comparing the predicted rollout to a ground-truth solution, the loss function measures how well the generated trajectory satisfies the governing physical laws. This is done by evaluating the residuals of the underlying partial differential equation (PDE), as well as the errors in satisfying the boundary conditions. These residuals are computed over the full predicted spatiotemporal grid, using finite difference approximations for both spatial and temporal derivatives. The total loss is then aggregated over all points and time steps in the trajectory.

The model parameters are updated through backpropagation through time (BPTT). By maintaining gradient flow through the full autoregressive sequence, the model learns a stable state transition function, or evolution operator, that captures long-range temporal dependencies and adheres to the physical constraints. This end-to-end training on physically constrained rollouts directly minimizes the one-step rollout error discussed in Theorem 3.4. As a result, the model mitigates error accumulation commonly found in non-autoregressive approaches and produces stable, accurate long-term predictions.

## C  ADDITIONAL DETAILS ON PDE BENCHMARK EXPERIMENT

### C.1  PDE SETUP

We evaluate PIANO on four canonical time-dependent partial differential equations (PDEs) that are standard benchmarks in the physics-informed machine learning literature. For each benchmark, we define the governing equation, the domain, the initial conditions (ICs), and the boundary conditions (BCs). The analytical solution for each PDE is provided for the purpose of evaluating model accuracy. Table 2 summarizes the unique numerical challenges posed by each equation, which test different aspects of a solver's stability and accuracy.

**1. Wave Equation:** The 1D wave equation models phenomena like vibrating strings and sound waves. It is a second-order hyperbolic PDE.

- **Equation**: $\frac{\partial^2 u}{\partial t^2} = c^2 \frac{\partial^2 u}{\partial x^2}$, with $c = 2.0$.

- **Domain**: $(x, t) \in [0, 1] \times [0, 1]$.
- **Initial Conditions**:
  - $u(x, 0) = \sin(\pi x) + 0.5 \sin(3\pi x)$.
  - $\frac{\partial u}{\partial t}(x, 0) = 0$.
- **Boundary Conditions**: $u(0, t) = 0$ and $u(1, t) = 0$ (Dirichlet).
- **Analytical Solution**: $u(x, t) = \sin(\pi x) \cos(2\pi t) + 0.5 \sin(3\pi x) \cos(6\pi t)$.

**2. Reaction Equation:** This equation models systems with nonlinear reaction dynamics, common in chemistry and biology. It is a first-order nonlinear PDE.

- **Equation**: $\frac{\partial u}{\partial t} = 5u(1 - u)$.
- **Domain**: $(x, t) \in [0, 2\pi] \times [0, 1]$.
- **Initial Condition**: $u(x, 0) = \exp\left(-\frac{(x-\pi)^2}{2(\pi/4)^2}\right)$.
- **Boundary Conditions**: $u(0, t) = u(2\pi, t)$ (Periodic).
- **Analytical Solution**: Let $h(x) = u(x, 0)$. Then $u(x, t) = \frac{h(x)e^{5t}}{h(x)e^{5t}+1-h(x)}$.

**3. Convection Equation:** A first-order hyperbolic PDE that models the transport of a quantity. It is known for being sensitive to numerical diffusion, where sharp features can be smoothed out by inaccurate solvers.

- **Equation**: $\frac{\partial u}{\partial t} + c\frac{\partial u}{\partial x} = 0$, with $c = 50$.
- **Domain**: $(x, t) \in [0, 2\pi] \times [0, 1]$.
- **Initial Condition**: $u(x, 0) = \sin(x)$.
- **Boundary Conditions**: $u(0, t) = u(2\pi, t)$ (Periodic).
- **Analytical Solution**: $u(x, t) = \sin(x - ct)$.

**4. Heat Equation:** The heat equation is a second-order parabolic PDE that describes heat distribution in a given region over time. It is a classic example of a diffusive system, which presents challenges related to numerical stiffness.

- **Equation**: $\frac{\partial u}{\partial t} = \alpha\frac{\partial^2 u}{\partial x^2}$, with $\alpha = 0.1$.
- **Domain**: $(x, t) \in [0, 1] \times [0, 1]$.
- **Initial Condition**: $u(x, 0) = \sin(\pi x)$.
- **Boundary Conditions**: $u(0, t) = 0$ and $u(1, t) = 0$ (Dirichlet).
- **Analytical Solution**: $u(x, t) = \sin(\pi x)e^{-\alpha\pi^2 t}$.

Table 2: Summary of numerical challenges presented by each PDE benchmark.

| PDE Benchmark | Primary Numerical Challenge |
| --- | --- |
| **Wave Equation** | Accurate handling of second-order temporal and spatial derivatives. Propagating wave solutions without numerical dispersion. |
| **Reaction Equation** | Modeling stiff nonlinear dynamics where solutions change rapidly. Capturing exponential growth accurately. |
| **Convection Equation** | Minimizing numerical diffusion to preserve the shape of the propagating wave. High wave speed ($c = 50$) makes it challenging. |
| **Heat Equation** | Handling diffusive processes and numerical stiffness, which can require very small time steps for traditional explicit solvers. |

## C.2 BASELINES

To rigorously evaluate **PIANO**, we benchmark it against a comprehensive suite of models that represent the cutting edge of physics-informed learning. These baselines were chosen to cover classical architectures, recent architectural innovations, and state-of-the-art sequential models, providing a multi-faceted comparison.

- **Canonical PINN**: This is the foundational framework introduced by Raissi et al. (2019). It employs a Multilayer Perceptron (MLP) as a universal function approximator that takes spatio-temporal coordinates $(x, t)$ as input and outputs the corresponding solution $u(x, t)$. The network is trained by minimizing a loss function composed of the PDE residuals, initial conditions, and boundary conditions, which are calculated using automatic differentiation. Its primary limitation, which motivates our work, is its point-wise prediction mechanism, which neglects temporal dependencies and often fails to propagate initial conditions accurately, leading to "failure modes" where the model converges to overly smooth or incorrect solutions.

- **QRes & FLS**: These models represent architectural improvements over the standard MLP. **QRes (Quadratic Residual Networks)** introduces quadratic residual connections to enhance the model's capacity for solving complex physics problems (Bu & Karpatne, 2021). **FLS (First-Layer Sine)** networks use a sinusoidal activation function in the first layer (Wong et al., 2022). This provides a strong inductive bias for learning periodic or high-frequency patterns, though its effectiveness can be limited to problems where such prior knowledge of the solution's behavior is applicable.

- **RoPINN (Region Optimized PINN)**: This framework addresses a fundamental deficiency in the standard PINN training paradigm (Wu et al., 2024). Instead of optimizing the loss on a finite set of scattered points, RoPINN extends the optimization to the continuous neighborhood regions around these points. This is achieved efficiently through a Monte Carlo sampling method within a "trust region" that is adaptively calibrated during training. This approach is designed to reduce generalization error and better satisfy high-order PDE constraints without requiring additional, costly gradient calculations.

- **KAN (Kolmogorov-Arnold Networks)**: Representing a recent breakthrough in neural network architecture, KANs are included as an advanced physics-informed backbone (Liu et al., 2025). They offer a powerful alternative to traditional MLPs and have demonstrated strong performance in several scientific machine learning tasks.

- **Sequential, Non-Autoregressive Models**: To highlight the specific advantage of PIANO's autoregressive nature, we compare against the most advanced sequential models.

  - **PINNsFormer** is a Transformer-based framework designed specifically to capture temporal dependencies (Zhao et al., 2024). Its key mechanism is the "Pseudo Sequence Generator," which transforms a point-wise input $(x, t)$ into a short temporal sequence, $\{[x, t], [x, t + \Delta t], ...\}$, which is then processed by a multi-head attention mechanism. While it processes information sequentially, its predictions at each time step are made independently of the model's own previous predictions, distinguishing it from PIANO's true autoregressive approach.

  - **PINNMamba** utilizes a State Space Model (SSM) to serve as a "continuous-discrete articulation," aiming to resolve the mismatch between continuous PDEs and discrete training points (Xu et al., 2025). It employs "sub-sequence modeling" and a contrastive alignment loss to combat the simplicity bias of neural networks and propagate initial conditions. Like PINNsFormer, it is a sequential model, but it is not autoregressive in the way PIANO is.

## C.3 COMPLEXITY ANALYSIS

We provide a complexity analysis to position PIANO relative to its baselines in terms of computational and memory overhead, with model details summarized in Table 3. Standard MLP-based PINNs serve as an efficient baseline, requiring approximately 1311 MiB of GPU memory. In contrast, advanced sequential models designed to capture temporal dependencies incur significantly greater costs. PINNsFormer, based on a Transformer architecture, exhibits quadratic computational

and memory complexity with respect to sequence length $M$, i.e., $\mathcal{O}(M^2)$, due to its self-attention mechanism. This makes it inherently inefficient for long sequences and results in a memory footprint of around $2827$ MiB, more than twice that of a standard PINN, along with nearly three times the computational cost. Similarly, PINNMamba adopts a state-space architecture that theoretically scales linearly as $\mathcal{O}(M)$, but in practice incurs a very high memory footprint of approximately $7899$ MiB and nearly seven times the time per iteration of a standard PINN. This overhead stems from its reliance on short sub-sequences (e.g., $k = 7$) and a complex sub-sequence contrastive alignment loss. PIANO, while also a sequential model, is explicitly designed for efficient long-range modeling. Like PINNMamba, it scales linearly with $M$ in theory. However, it avoids the need for sub-sequence processing and specialized losses. Instead, it processes the entire temporal rollout as a single long sequence (e.g., $M = 200$ in our experiments) using a streamlined autoregressive architecture. As shown in Table 3, this results in a substantially lower memory usage of approximately $800$ MiB, which is lower than both PINNMamba and even the MLP-based PINN baseline. While PIANO's autoregressive nature introduces a modest increase in computational cost per iteration compared to MLPs, it remains comparable to other sequential baselines without incurring their prohibitive memory overhead. Overall, PIANO achieves an effective balance between expressive power, scalability, and resource efficiency, making it well suited for simulating long-horizon dynamical systems.

### C.4 HYPERPARAMETERS AND EXPERIMENTAL DETAILS

To ensure full reproducibility of our results, this section provides comprehensive information on the evaluation metrices, experimental setup, training configurations, and specific hyperparameters used for both our proposed model, **PIANO**, and all baseline models.

### C.4.1 EXPERIMENTAL SETUP

All experiments were conducted on a workstation equipped with NVIDIA A100 GPUs with GPU memory usage of $\sim 800$MiB. Our codebase is implemented in PyTorch. For the PDE benchmark tasks, we trained all models on a discretized spatio-temporal grid of $200 \times 200$ collocation points. For evaluation and visualization, a distinct grid of $198 \times 198$ points was used. Although PIANO does not involve significant sources of randomness that affect performance, experiments are repeated with seeds 0 through 9 for multiple-run evaluations. For single-run experiments, seed 0 is used. We provide the source code in the attached zip file with software libraries and their versions in the .toml file. The source code will also be made public upon acceptance.

### C.4.2 TRAINING CONFIGURATION

PIANO was trained for 100K iterations using the AdamW optimizer with a learning rate of $3 \times 10^{-4}$ and a weight decay of $10^{-4}$. A cosine annealing learning rate scheduler was employed to gradually reduce the learning rate over the course of training. To stabilize optimization, gradient norms were clipped to a maximum of 1.0. Model weights were initialized using the Xavier uniform initialization strategy, and the best-performing model (based on training loss) was checkpointed and used for final evaluation. Throughout training, intermediate predictions were saved at regular intervals to support qualitative analysis. All spatial and temporal derivatives required for the PDE loss were approximated using second-order finite difference schemes applied to the model's predicted solution grid. When enabled, Weights & Biases (WandB) was used to log training losses, runtime, and gradient diagnostics. All models were trained on a uniform spatiotemporal grid, with analytically defined initial conditions specific to each PDE benchmark.

### C.4.3 MODEL ARCHITECTURE DETAILS

Our proposed model, PIANO, is implemented as a State-Space Model (SSM) architecture. The key components and hyperparameters are detailed below, followed by those of the primary baselines.

- **PIANO (Ours)**: Our model, referred to as 'ssm' in our experiments, is an autoregressive state-space model.

Table 3: Architectural hyperparameters for PIANO and baseline models used in the PDE benchmarks.

| Model | Hyperparameter | Value |
|---|---|---|
| **PINN / FLS** | Hidden Layers | 4 |
| | Hidden Size | 512 |
| | Total Parameters | $\sim$527k |
| | GPU Memory | $\sim$1311MiB |
| **QRes** | Hidden Layers | 4 |
| | Hidden Size | 256 |
| | Total Parameters | $\sim$397k |
| **PINNsFormer** | Sequence Length ($k$) | 5 |
| | Time Step ($\Delta t$) | $10^{-4}$ |
| | Number of Encoders/Decoders | 1 |
| | Embedding Size | 32 |
| | Attention Heads | 2 |
| | Hidden Size | 512 |
| | Total Parameters | $\sim$454k |
| | GPU Memory | $\sim$2827MiB |
| **PINNMamba** | Sequence Length ($k$) | 7 |
| | Time Step ($\Delta t$) | $10^{-2}$ |
| | Number of Encoders | 1 |
| | Embedding Size | 32 |
| | Total Parameters | $\sim$286k |
| | GPU Memory | $\sim$7899MiB |
| **PIANO (Ours)** | State Dimension ($k$) | 256 |
| | Total Parameters | $\sim$330k |
| | GPU Memory | $\sim$800MiB |

- **Input**: At each time step $t$, the input vector is a concatenation of the spatial coordinate $x_t$, the temporal coordinate $t_t$, and the model's own prediction from the previous step, $u_{t-1}$.
- **Embedding Network**: A linear layer maps the concatenated input vector to a hidden state dimension of 256.
- **State Transition Network**: The core of our model consists of learnable matrices (**A, B, C, D**) that govern the state-space dynamics. The hidden state dimension is 256, and we use a SiLU (silu) activation function with Layer Normalization for stability.
- **PDE Probe**: A 2-layer MLP with a hidden dimension of 256 and a SiLU activation decodes the hidden state into the final solution at each time step.

Table 3 summarizes the architectural hyperparameters for PIANO and the primary baseline models to ensure a fair comparison in terms of model capacity. We aimed to keep the total number of trainable parameters roughly comparable across the different architectures.

### C.4.4 EVALUATION METRICS

To quantitatively assess the accuracy of our model and the baselines, we employ two primary evaluation metrics: the relative Mean Absolute Error (rMAE) and the relative Root Mean Squared Error (rRMSE). These metrics are standard and widely adopted throughout the PINN research literature, ensuring our results are comparable with prior and future work (Xu et al., 2025; Wu et al., 2024; Zhao et al., 2024).

We use relative error metrics instead of absolute ones (e.g., MAE or RMSE) because they provide a scale-invariant measure of performance. Absolute errors are dependent on the magnitude of the PDE's solution; a physically correct model for a high-magnitude field (like pressure) could have a large absolute error, while a poor model for a normalized field (like concentration) could have

Table 4: Mean ± standard deviation of rMAE over 10 runs of PIANO and the second-best baseline for each PDE. PIANO consistently achieves both lower error and lower variance.

| Model | Wave | Reaction | Convection | Heat |
|---|---|---|---|---|
| Second best | $0.0452 \pm 0.0045$ | $0.0183 \pm 0.0016$ | $0.0395 \pm 0.0032$ | $0.0821 \pm 0.0061$ |
| PIANO (ours) | $\mathbf{0.0057 \pm 0.0004}$ | $\mathbf{0.0001 \pm 0.0000}$ | $\mathbf{0.0032 \pm 0.0003}$ | $\mathbf{0.0000 \pm 0.0000}$ |

Table 5: Mean ± standard deviation of rRMSE over 10 runs of PIANO and the second-best baseline for each PDE. PIANO consistently achieves both lower error and lower variance.

| Model | Wave | Reaction | Convection | Heat |
|---|---|---|---|---|
| Second best | $0.0487 \pm 0.0048$ | $0.0246 \pm 0.0021$ | $0.0432 \pm 0.0035$ | $0.0894 \pm 0.0065$ |
| PIANO (ours) | $\mathbf{0.0059 \pm 0.0005}$ | $\mathbf{0.0008 \pm 0.0001}$ | $\mathbf{0.0104 \pm 0.0007}$ | $\mathbf{0.0002 \pm 0.0000}$ |

a small one. Relative errors normalize the error by the magnitude of the true solution, providing a dimensionless percentage that is directly comparable across different PDEs, scales, and physical units. This is essential for a robust and generalizable evaluation.

Given a set of $N$ test points, the model's prediction $\hat{u}(x_n, t_n)$, and the ground truth analytical solution $u(x_n, t_n)$, the metrics are formulated as follows:

$$\text{rMAE} = \frac{\sum_{n=1}^{N} |\hat{u}(x_n, t_n) - u(x_n, t_n)|}{\sum_{n=1}^{N} |u(x_n, t_n)|} \tag{23}$$

$$\text{rRMSE} = \sqrt{\frac{\sum_{n=1}^{N} |\hat{u}(x_n, t_n) - u(x_n, t_n)|^2}{\sum_{n=1}^{N} |u(x_n, t_n)|^2}} \tag{24}$$

### C.5 SIGNIFICANCE ANALYSIS

Tables 4 and 5 present the mean and standard deviation of relative errors across 10 independent runs on the PDE benchmark. In both rMAE and rRMSE, PIANO consistently achieves substantially lower error and variance compared to the second-best baseline. A two-tailed $t$-test with $p < 0.05$ confirms that the improvements are statistically significant across all PDEs, establishing PIANO as both more accurate and more stable.

### C.6 ADDITIONAL QUALITATIVE RESULTS

### C.6.1 TRAINING DYNAMICS

To provide further insight into the autoregressive behavior of PIANO, we visualize its training dynamics across different PDEs. Figures 7–10 show spacetime predictions $u(x, t)$ at various stages of training (5%, 25%, 50%, and 100%), illustrating how the solution progressively improves. These visualizations highlight how the model gradually reconstructs the full trajectory by propagating the known initial condition forward in time using its own predictions.

For the transport-dominated problems like the Convection equation (main paper Figure 3) highlight PIANO's ability to preserve sharp wavefronts over long horizons. This demonstrates the model's robustness in avoiding numerical diffusion, an issue that often affects other PINN-based methods.

The Wave equation (Figure 7) requires learning second-order oscillatory dynamics, making convergence more gradual. Early in training, the model underfits both amplitude and phase. However, by the end of training, PIANO successfully recovers the full oscillatory structure, without suffering from phase drift or artificial dispersion. The solution is slowly and correctly propagated from the initial stages to the later.

For the Heat equation (Figure 9), PIANO converges extremely rapidly. As a diffusion-dominated PDE, the solution is smooth and stable, allowing the model to reach near-zero error by 50% of training.

The Reaction equation, shown in Figures 8 and 10, provides a particularly informative case for analyzing autoregressive propagation. Although convergence is still relatively fast, we include an extended grid of training snapshots to illustrate how the model handles the nonlinearity and exponential growth in the solution. Early predictions accurately reconstruct the region near $t = 0$, where the initial condition serves as an anchor. As training progresses, the predicted solution propagates deeper into the temporal domain, revealing how PIANO gradually builds up the full dynamics through stable, recursive conditioning. This dense visualization helps expose the internal mechanics of the autoregressive learning process.

Overall, these qualitative results confirm that PIANO maintains stable and physically consistent rollout behavior across a wide range of PDE types, from diffusive to oscillatory, and from linear to nonlinear dynamics.

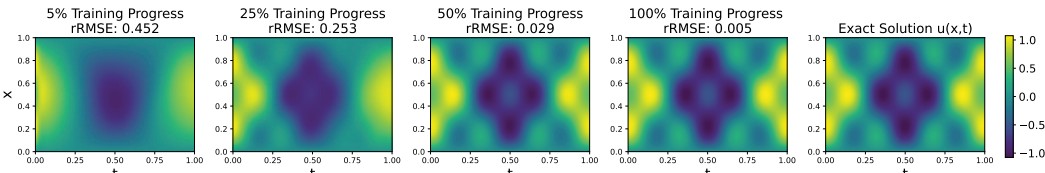

Figure 7: Training progression of PIANO on the wave equation. Each subplot shows the predicted spacetime solution $u(x, t)$ at different training stages, with the exact solution on the far right. The model progressively learns the oscillatory structure of the solution, despite the complexity of second-order dynamics. Accurate predictions emerge early near the initial condition and gradually propagate forward in time through the autoregressive rollout. By 100% progress (rRMSE: 0.005), the model closely matches the exact solution across the domain.

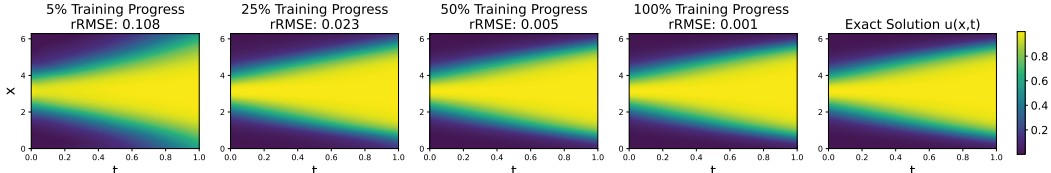

Figure 8: Training progression of PIANO on a reaction equation. Each subplot shows the predicted spacetime solution $u(x, t)$ at different training stages, with the exact solution on the far right. At 5% progress, the model captures the coarse global structure but underestimates the peak and shows boundary errors. As training proceeds, accurate predictions emerge first near the initial state and gradually propagate forward in time through the autoregressive rollout. By 100% progress (rRMSE: 0.0008), the model closely matches the exact solution. Extended dynamics are shown in Figure 10.

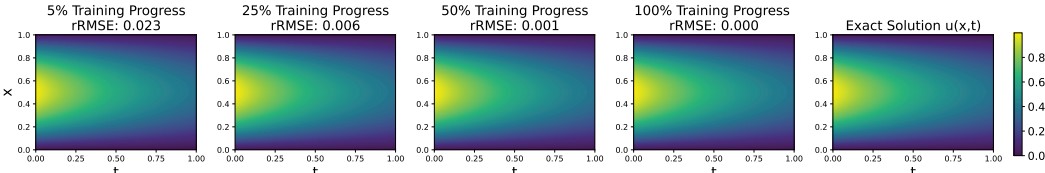

Figure 9: Training progression of PIANO on the heat equation. Each subplot shows the predicted spacetime solution $u(x, t)$ at different training stages, with the exact solution on the far right. Due to the diffusive nature of the equation, the model converges rapidly, capturing the correct solution structure early in training. Accurate dynamics propagate smoothly from the initial condition, with near-perfect agreement reached by 50% progress (rRMSE: 0.001).

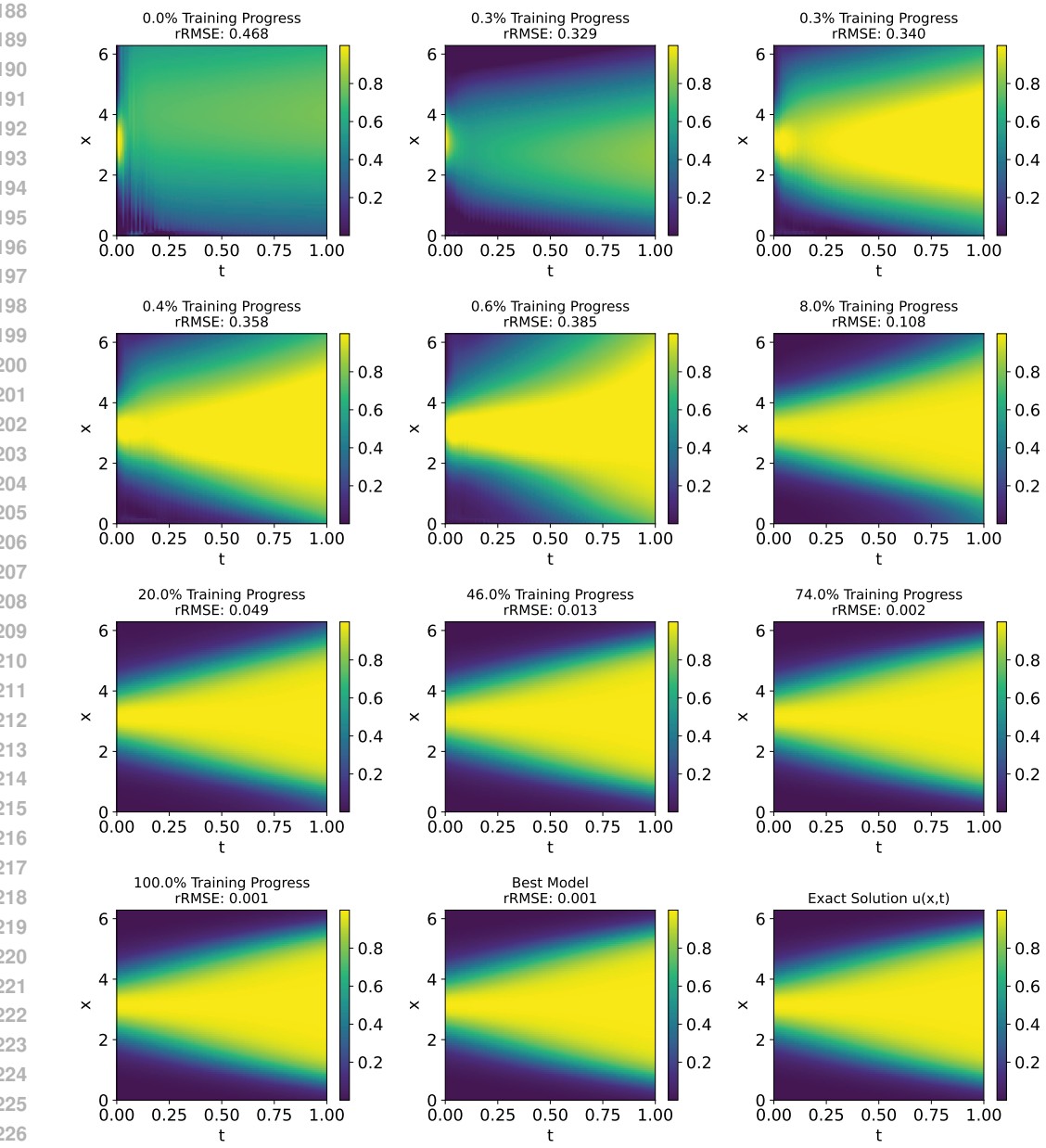

Figure 10: Extended training dynamics of PIANO on the reaction equation. This snapshot grid shows how accurate solution structures first emerge near the initial condition and progressively propagate forward in time as training advances. The gradual refinement across training stages illustrates the model's ability to learn stable temporal evolution through its autoregressive rollout.

### C.6.2 TEMPORAL PROFILE

To further analyze the temporal behavior learned by PIANO, we visualize in Figures 11 and 12 the predicted time profiles $u(x_i, t)$ across several fixed spatial locations $x_i$, at different stages of training for the Wave and Convection equations, respectively. Each subplot corresponds to a particular training progress percentage, with colored curves representing different spatial points.

For the Wave equation, the model initially struggles to capture the high-frequency oscillations, exhibiting distorted amplitudes and poor phase alignment. As training progresses, PIANO gradually learns to reconstruct both the amplitude and frequency content of the wave. Notably, improvements

emerge near the initial time and propagate forward, consistent with PIANO's autoregressive rollout structure. This recursive conditioning enables the model to build the solution step by step, avoiding phase drift and dispersion commonly seen in non-autoregressive methods. By 100% training, the predicted oscillations closely match the ground truth in amplitude, frequency, and phase across all $x_i$, indicating that PIANO has successfully learned the global wave dynamics in a stable and physically consistent manner.

The Convection equation presents a different challenge due to its transport-dominated nature. In the early stages of training, predictions are only accurate near the initial time, while further regions suffer from amplitude decay and misaligned phase. However, as training advances, PIANO progressively learns to propagate the sharp waveform forward in time. By conditioning each step on its own prior predictions, the model gradually sharpens the solution and aligns the oscillatory phase across all spatial locations. By the final stages of training, PIANO maintains the structure and timing of the waveform without numerical diffusion, demonstrating its robustness in capturing transport dynamics through stable temporal propagation.

These visualizations confirm that PIANO not only handles oscillatory PDEs like the Wave equation but also excels in transport-heavy regimes like Convection, leveraging its autoregressive architecture to deliver stable and accurate long-term predictions.

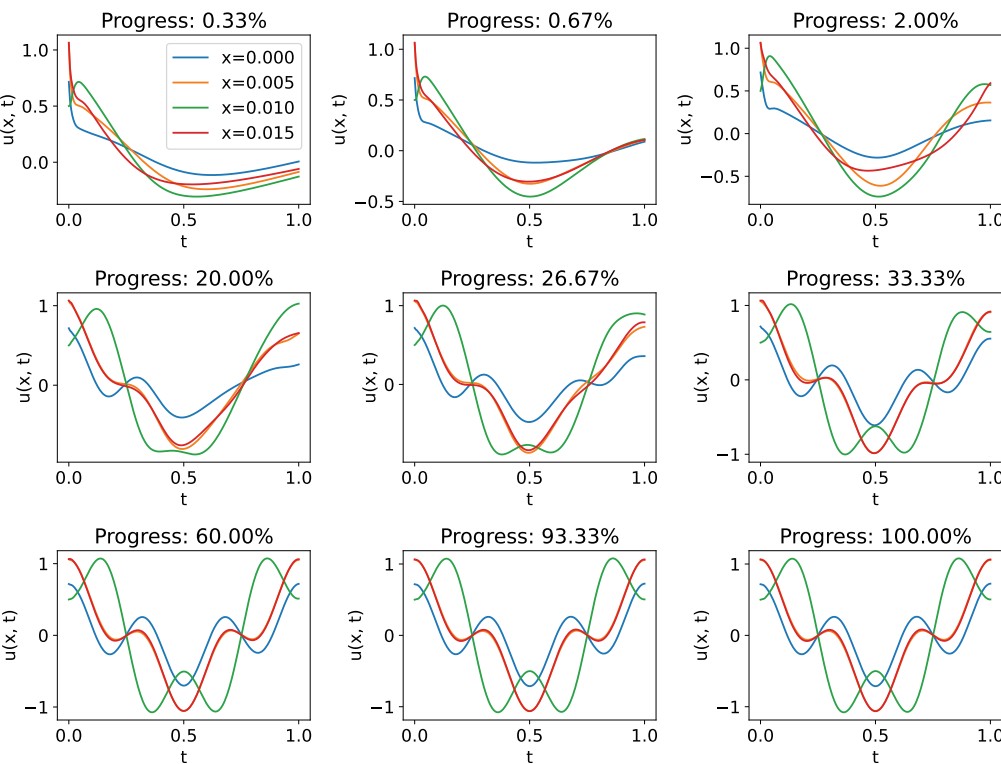

Figure 11: Temporal profile of the Wave equation at various training stages. Each subplot shows the predicted time evolution $u(x_i, t)$ for multiple spatial points $x_i$, plotted as separate curves. Early in training (top row), the model captures only coarse low-frequency behavior and underestimates amplitude. As training progresses, PIANO improves its ability to preserve the phase and frequency content of oscillations across all spatial locations. This gradual sharpening of periodic structure demonstrates the autoregressive nature of the model: accurate dynamics emerge near the initial time and propagate forward as the model recursively builds on its own predictions. By the final stages, the predicted waveforms closely match in both phase and amplitude across the entire domain, confirming stable temporal learning.

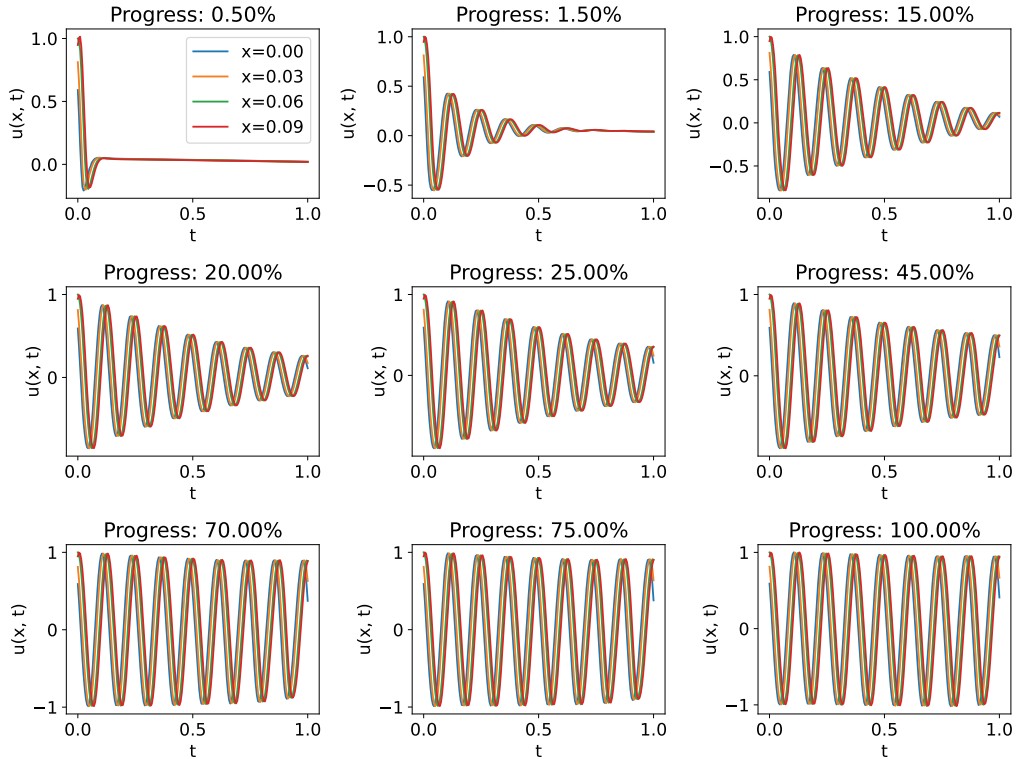

Figure 12: Temporal profiles of the Convection equation at different training stages. Each subplot shows the predicted evolution $u(x_i, t)$ over time for several fixed spatial locations $x_i$. Early predictions are accurate only near the initial time, but fail to preserve the wave shape across the domain. As training progresses, PIANO gradually learns to propagate the waveform forward in time, maintaining sharpness and phase alignment. By 100% training progress, the model stably reproduces the full dynamics, demonstrating robust temporal consistency under transport-dominated behavior.

### C.7 GUIDANCE ON HYPERPARAMETERS AND ABLATION

**Guidance on hyperparameters** The architectural hyperparameters in PIANO, namely the state dimension $(k)$ and the number of temporal rollout steps $(M)$, are standard and do not introduce novel tuning complexities. The value for $M$ is determined by the temporal resolution of the training grid (e.g., for a $200 \times 200$ spatio-temporal grid, $M = 200$). To provide clear guidance for practitioners, we perform an empirical sensitivity analysis on these key parameters. We investigate the impact of the state dimension and the training grid resolution on the model's accuracy for the Reaction equation.

The results are presented in Table 6 and Table 7. We observe two clear trends:

- Increasing the state dimension from 32 to 256 substantially reduces prediction error, demonstrating the benefit of higher model capacity for capturing the underlying dynamics of the PDE.

- Increasing the training grid resolution from $50 \times 50$ to $200 \times 200$ consistently improves performance, as a denser sampling of collocation points provides stronger and more complete physical constraints during training.

Based on these findings, we used a state dimension of $k = 256$ and a grid size of $200 \times 200$ for our main experiments to ensure the highest accuracy. We do not go further as the error is close to zero.

**Ablation Studies** We perform ablation studies on the 1D Reaction equation to evaluate the contributions of PIANO's two key components: its high-order finite difference scheme for computing

Table 6: Sensitivity analysis of PIANO with respect to the state dimension ($k$) for the Reaction equation. The model is trained on a $200 \times 200$ grid. Errors decrease as the state dimension increases.

| State Dimension ($k$) | rMAE ($\downarrow$) | rRMSE ($\downarrow$) |
|:---:|:---:|:---:|
| 32 | 0.0145 | 0.0284 |
| 64 | 0.0025 | 0.0047 |
| 128 | 0.0016 | 0.0038 |
| **256** | **0.0001** | **0.0008** |

Table 7: Sensitivity analysis of PIANO with respect to the training grid resolution for the Reaction equation. The model uses a fixed state dimension of $k = 256$. Performance improves with a finer grid.

| Grid Size | rMAE ($\downarrow$) | rRMSE ($\downarrow$) |
|:---:|:---:|:---:|
| $50 \times 50$ | 0.0017 | 0.0047 |
| $100 \times 100$ | 0.0005 | 0.0009 |
| $200 \times 200$ | **0.0001** | **0.0008** |

derivatives and the autoregressive architecture. All reported results are averaged over ten independent runs to ensure statistical reliability. Table 8 summarizes the findings.

### C.7.1 FINITE DIFFERENCE SCHEME

PIANO employs finite differences (FD) instead of automatic differentiation (AD) to approximate PDE derivatives. This choice provides stable gradients, lower memory requirements, and allows training with modern first-order optimizers such as AdamW (Kingma & Ba, 2014). In contrast, AD-based PINNs typically rely on quasi-second-order optimizers such as L-BFGS, which are less scalable and poorly suited for stochastic mini-batch training.

To assess the importance of derivative precision, we compare a second-order FD scheme against a first-order version. The second-order scheme achieves an rRMSE of $0.0008$, while the first-order scheme yields $0.0174$. This difference is statistically significant (two-tailed t-test, $p < 0.05$), demonstrating that higher-order derivative approximations are essential for accurate physics-informed training. Notably, even the first-order version of PIANO performs better than most baselines in Table 1, underscoring the robustness of the approach.

### C.7.2 AUTOREGRESSIVE BACKBONE

Using the validated second-order FD scheme, we now examine PIANO's autoregressive architecture. A non-autoregressive baseline ("PIANO (Non-AR)"), which functions like a standard PINN, performs poorly with an rRMSE of $0.8010$, confirming that autoregressive formulation is critical. Introducing progressively stronger recurrent backbones significantly improves performance: an MLP-based variant reaches $0.0502$, a GRU-based model achieves $0.0061$, and the full state-space architecture attains the lowest error of $0.0008$. All improvements are statistically significant ($p < 0.05$).

These results demonstrate that PIANO's autoregressive design is fundamental to its accuracy and stability. Even a basic autoregressive MLP variant dramatically outperforms the non-autoregressive baseline, and the full state-space backbone achieves near-zero error, validating its role as the most effective temporal modeling choice.

## D WEATHER FORECASTING

### D.1 EXTENDED SETUP AND IMPLEMENTATION

**Background** Weather forecasting has traditionally been dominated by numerical simulations of complex atmospheric physics. Although powerful, these methods are computationally demanding. Recently, deep learning models have emerged as a promising alternative, yet they often function

Table 8: Ablation study on the 1D Reaction equation (mean over ten runs). Results isolate the contributions of PIANO's autoregressive backbone and the precision of the finite difference scheme

| Method | rRMSE ($\downarrow$) |
|---|---|
| **Finite Difference Schemes** | |
| First Order Accurate | $0.0174 \pm 0.0195$ |
| Second Order Accurate | $0.0008 \pm 0.0001$ |
| **Autoregressive Backbone** | |
| PIANO (Non-AR) | $0.8010 \pm 0.1293$ |
| PIANO (MLP) | $0.0502 \pm 0.0017$ |
| PIANO (GRU) | $0.0061 \pm 0.0009$ |
| PIANO (SSM) | $0.0008 \pm 0.0001$ |

as a "black-box" that neglect the underlying physical principles. A more robust approach involves integrating physical laws with deep learning approaches. ClimODE (Verma et al., 2024) is a recent model that successfully applies the physics-informed strategy to weather forecasting. It is built on a core principle from statistical mechanics: weather evolution can be described as a continuous-time advection process, which models the spatial movement and redistribution of quantities like temperature and pressure. By framing the problem as a neural Ordinary Differential Equation (ODE) that adheres to the advection equation, ClimODE enforces value-conserving dynamics, a strong inductive bias that leads to more stable and physically plausible forecasts. With PIANO, we build on the ClimODE framework by introducing an autoregressive training scheme to further enhance predictive accuracy.

**Setup** Weather forecasting involves predicting the evolution of key atmospheric variables such as atmospheric temperature (`t`), surface temperature (`t2m`), horizontal wind components (`u10`, `v10`), and geopotential (`z`). We adopt the physics-informed framework of ClimODE (Verma et al., 2024), which models weather evolution as a continuous-time process governed by a system of neural ODEs. This system jointly evolves the weather state, denoted by $u(t)$, and a corresponding velocity field, $v(t)$.

The ODE system has two components. The first governs the rate of change of the weather state, $\dot{u}$, and is constrained by the physical advection equation, which ensures that quantities are transported and conserved according to physical principles. The second component governs the rate of change of the velocity field, $\dot{v}$, which is learned by a neural network, $f_\theta$. This network takes as input the current state $u(\tau)$, its spatial gradient $\nabla u(\tau)$, the velocity $v(\tau)$, and spatiotemporal embeddings $\psi$ to determine the acceleration of the flow.

In PIANO, we use this same physics-informed ODE structure but introduce an autoregressive training strategy with teacher forcing to reduce error accumulation. Instead of forecasting the entire trajectory in one step, the model is conditioned on the ground truth from the previous time point. The revised forecast equation for a single time step from $t_i$ to $t_j$ is given by:

$$\begin{bmatrix} \hat{u}(t_j) \\ \hat{v}(t_j) \end{bmatrix} = \begin{bmatrix} y_i \\ v(t_i) \end{bmatrix} + \int_{t_i}^{t_j} \begin{bmatrix} -\nabla \cdot (\hat{u}(\tau)\hat{v}(\tau)) \\ f_\theta(\hat{u}(\tau), \nabla\hat{u}(\tau), \hat{v}(\tau), \psi) \end{bmatrix} d\tau,$$

where $y_i$ denotes the observed ground truth state at time $t_i$, $v(t_i)$ is the inferred velocity at that time, and $\nabla\cdot$ is the spatial divergence operator.

We evaluate PIANO on the ERA5 dataset Rasp et al. (2020), a benchmark for global weather forecasting providing 6-hourly reanalysis data at $5.625°$ resolution for five variables: `t`, `t2m`, `u10`, `v10`, and `z`. We compare against several state-of-the-art baselines including Neural ODE (NODE) Verma et al. (2024), FCN Pathak et al. (2022), ClimaX Nguyen et al. (2023), and the original ClimODE Verma et al. (2024). As a reference, we also report results for the gold-standard Integrated Forecasting System (IFS) ECMWF (2023) which is one of the most advanced global physics simulation model and has high computational demands. Performance is evaluated using two standard metrics: root mean square error (RMSE) and anomaly correlation coefficient (ACC). RMSE quantifies the absolute prediction error, while ACC measures the correlation between predicted and

observed anomalies, capturing the directional accuracy. Both metrics are latitude-weighted to reflect the spherical geometry of the Earth.

**Implementation**   Our experimental framework directly mirrors the setup used by ClimODE to ensure a fair comparison. The primary task is forecasting future atmospheric states based on an initial state, with lead times ranging from 6 to 36 hours. The model is implemented in PyTorch. The underlying system of ODEs is solved using the Euler method with a time resolution of 1 hour, managed by the 'torchdiffeq' library. All experiments are conducted on a single NVIDIA A100 GPU with 40GB of memory. The model is trained for 300 epochs using a batch size of 8. The learning rate is managed by a Cosine Annealing scheduler.

**Evaluation Metrics**   We assess model performance using two standard meteorological metrics: latitude-weighted Root Mean Squared Error (RMSE) and Anomaly Correlation Coefficient (ACC), computed after de-normalizing the predictions.

$$\text{RMSE} = \frac{1}{N} \sum_{t=1}^{N} \left[ \frac{1}{HW} \sum_{h=1}^{H} \sum_{w=1}^{W} \alpha(h)(y_{thw} - u_{thw})^2 \right]^{1/2} \tag{25}$$

$$\text{ACC} = \frac{\sum_{t,h,w} \alpha(h)\tilde{y}_{thw}\tilde{u}_{thw}}{\sqrt{\sum_{t,h,w} \alpha(h)\tilde{y}_{thw}^2 \sum_{t,h,w} \alpha(h)\tilde{u}_{thw}^2}} \tag{26}$$

Here, $y_{thw}$ and $u_{thw}$ denote the ground truth and model prediction at time $t$, latitude index $h$, and longitude index $w$, respectively. The term $\alpha(h) = \cos(h) / \frac{1}{H} \sum_{h'} \cos(h')$ represents the normalized latitude weight, accounting for the area distortion in latitude-longitude grids due to Earth's curvature.

The anomalies are computed by subtracting the empirical mean:

$$\tilde{y}_{thw} = y_{thw} - C, \quad \tilde{u}_{thw} = u_{thw} - C,$$

where $C = \frac{1}{N} \sum_t y_{thw}$.

ACC measures the correlation between predicted and true anomalies. Higher ACC indicates better skill in capturing deviations from climatological means. Latitude-weighted RMSE evaluates the spatial accuracy of forecasts while correcting for latitudinal area distortion. Lower RMSE and higher ACC both indicate better forecasting performance.

**Dataset and Preprocessing**   We use the ERA5 dataset, as preprocessed for the WeatherBench benchmark. The data is provided at a 5.625° spatial resolution with a 6-hour time increment. Our experiments focus on five key variables: 2-meter temperature (t2m), temperature at 850 hPa (t), geopotential at 500 hPa (z), and the 10-meter U and V wind components (u10, v10). All variables are normalized to a [0, 1] range using min-max scaling. The dataset is partitioned by year, with 2006-2015 used for training, 2016 for validation, and 2017-2018 for testing.

## D.2   RESULTS

We evaluate PIANO's ability to forecast global weather variables using the ERA5 dataset. Figure 13 provides a visual analysis of PIANO's probabilistic predictions (extended from the ClimODE framework) at a fixed forecast time (2017-01-01T12:00) across five key atmospheric variables: geopotential height at 500 hPa (z), temperature at 850 hPa (t), 2-meter surface temperature (t2m), and the 10-meter U and V wind components (u10, v10). Each row corresponds to a variable, while the columns show the predicted mean ($\mu$), upper bound ($\mu + \sigma$), predicted standard deviation ($\sigma$), and pointwise error. These results demonstrate that PIANO not only captures the spatial structure of each variable, but also quantifies predictive uncertainty effectively, with visually low error and consistent uncertainty estimates across regions.

All uncertainty visualizations (e.g., predicted $\sigma$ maps) and uncertainty-based metrics (e.g., CRPS) shown in this paper are taken from the ClimODE's probabilistic emission model, specifically, the Gaussian observation model

$$y_k(x,t) \sim \mathcal{N}\big(u_k(x,t) + \mu_k(x,t),\, \sigma_k^2(x,t)\big),$$

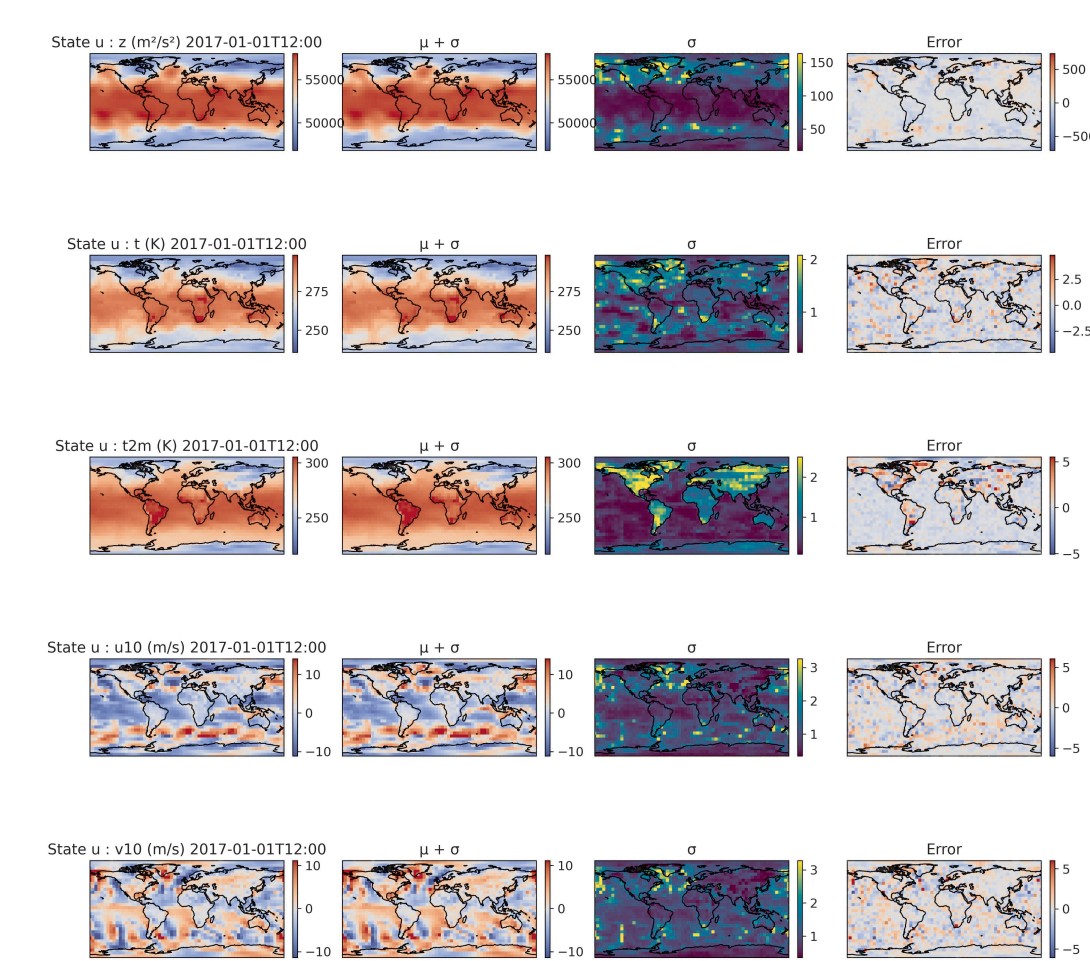

Figure 13: Visualization of PIANO's forecasting capabilities for five atmospheric state variables on 2017-01-01 at 12:00. Each row corresponds to a different variable: geopotential (z), atmospheric temperature (t), 2-meter surface temperature (t2m), and the 10-meter U-wind (u10) and V-wind (v10) components. The columns show (from left to right): the predicted mean $\mu$, upper bound $\mu + \sigma$, predicted standard deviation $\sigma$, and prediction error (difference from ground truth). PIANO captures spatial structure and uncertainty across variables, with low errors. The errors are more pronounced where PIANO already suggests uncertainty ($\sigma$).

with learnable bias $\mu_k$ and variance $\sigma_k^2$ trained via negative log-likelihood. As we extend ClimODE with PIANO the probabilistic emission model is naturally extended.

Quantitative results are summarized in Table 9, where we report latitude-weighted RMSE and Anomaly Correlation Coefficient (ACC) at multiple forecast lead times, comparing PIANO against several strong neural and numerical baselines, including NODE, ClimaX, FCN, IFS, and ClimODE. Across all variables and lead times, PIANO achieves state-of-the-art performance, often outperforming neural baselines by a significant margin and in some cases approaching the accuracy of IFS. The model shows particularly strong gains in mid-range horizons (12–24 hours), maintaining high correlation and low error while producing calibrated uncertainty estimates. These results highlight the benefit of PIANO's autoregressive, physics-informed structure for long-range, high-resolution weather modeling.

Table 9: Latitude weighted RMSE(↓) and Anomaly Correlation Coefficient (ACC(↑)) comparison with baselines on global forecasting on the ERA5 dataset. PIANO generally outperforms all the neural baselines.

| Variable | Lead-Time (hrs) | RMSE(↓) | | | | | | ACC(↑) | | | | | |
|---|---|---|---|---|---|---|---|---|---|---|---|---|---|
| | | NODE | ClimaX | FCN | IFS | ClimODE | PIANO (Ours) | NODE | ClimaX | FCN | IFS | ClimODE | PIANO (Ours) |
| z | 6 | 300.64 | 247.5 | 149.4 | 26.9 | $102.9 \pm 9.3$ | $69.07 \pm 4.99$ | 0.96 | 0.97 | 0.99 | 1.00 | 0.99 | 1.00 |
| | 12 | 460.23 | 265.4 | 217.8 | (N/A) | $134.8 \pm 12.3$ | $109.07 \pm 8.30$ | 0.88 | 0.96 | 0.99 | (N/A) | 0.99 | 0.99 |
| | 18 | 627.65 | 319.8 | 275.0 | (N/A) | $162.7 \pm 14.4$ | $145.99 \pm 11.95$ | 0.79 | 0.95 | 0.99 | (N/A) | 0.98 | 0.99 |
| | 24 | 877.82 | 364.9 | 333.0 | 51.0 | $193.4 \pm 16.3$ | $185.22 \pm 15.91$ | 0.70 | 0.93 | 0.99 | 1.00 | 0.98 | 0.98 |
| | 36 | 1028.20 | 455.0 | 449.0 | (N/A) | $259.6 \pm 22.3$ | $263.44 \pm 22.96$ | 0.55 | 0.89 | 0.99 | (N/A) | 0.96 | 0.97 |
| t | 6 | 1.82 | 1.64 | 1.18 | 0.69 | $1.16 \pm 0.06$ | $0.92 \pm 0.04$ | 0.94 | 0.94 | 0.99 | 0.99 | 0.97 | 0.98 |
| | 12 | 2.32 | 1.77 | 1.47 | (N/A) | $1.32 \pm 0.13$ | $1.16 \pm 0.05$ | 0.85 | 0.93 | 0.99 | (N/A) | 0.96 | 0.97 |
| | 18 | 2.93 | 1.93 | 1.65 | (N/A) | $1.47 \pm 0.16$ | $1.32 \pm 0.06$ | 0.77 | 0.92 | 0.99 | (N/A) | 0.96 | 0.96 |
| | 24 | 3.35 | 2.17 | 1.83 | 0.87 | $1.55 \pm 0.18$ | $1.48 \pm 0.07$ | 0.72 | 0.90 | 0.99 | 0.99 | 0.95 | 0.96 |
| | 36 | 4.13 | 2.49 | 2.21 | (N/A) | $1.75 \pm 0.26$ | $1.76 \pm 0.09$ | 0.58 | 0.86 | 0.99 | (N/A) | 0.94 | 0.94 |
| t2m | 6 | 2.72 | 2.02 | 1.28 | 0.97 | $1.21 \pm 0.09$ | $1.01 \pm 0.05$ | 0.82 | 0.92 | 0.99 | 0.99 | 0.97 | 0.98 |
| | 12 | 3.16 | 2.26 | 1.48 | (N/A) | $1.45 \pm 0.10$ | $1.20 \pm 0.09$ | 0.68 | 0.90 | 0.99 | (N/A) | 0.96 | 0.97 |
| | 18 | 3.45 | 2.45 | 1.61 | (N/A) | $1.43 \pm 0.09$ | $1.29 \pm 0.08$ | 0.69 | 0.88 | 0.99 | (N/A) | 0.96 | 0.97 |
| | 24 | 3.86 | 2.37 | 1.68 | 1.02 | $1.40 \pm 0.09$ | $1.42 \pm 0.10$ | 0.79 | 0.89 | 0.99 | 0.99 | 0.96 | 0.96 |
| | 36 | 4.17 | 2.87 | 1.90 | (N/A) | $1.70 \pm 0.15$ | $1.68 \pm 0.15$ | 0.49 | 0.83 | 0.99 | (N/A) | 0.94 | 0.94 |
| u10 | 6 | 2.30 | 1.58 | 1.47 | 0.80 | $1.41 \pm 0.07$ | $1.24 \pm 0.06$ | 0.85 | 0.92 | 0.95 | 0.98 | 0.91 | 0.95 |
| | 12 | 3.13 | 1.96 | 1.89 | (N/A) | $1.81 \pm 0.09$ | $1.53 \pm 0.07$ | 0.70 | 0.88 | 0.93 | (N/A) | 0.89 | 0.93 |
| | 18 | 3.41 | 2.24 | 2.05 | (N/A) | $1.97 \pm 0.11$ | $1.74 \pm 0.07$ | 0.58 | 0.84 | 0.91 | (N/A) | 0.88 | 0.91 |
| | 24 | 4.10 | 2.49 | 2.33 | 1.11 | $2.01 \pm 0.10$ | $1.96 \pm 0.09$ | 0.50 | 0.80 | 0.89 | 0.97 | 0.87 | 0.88 |
| | 36 | 4.68 | 2.98 | 2.87 | (N/A) | $2.25 \pm 0.18$ | $2.35 \pm 0.12$ | 0.35 | 0.69 | 0.85 | (N/A) | 0.83 | 0.83 |
| v10 | 6 | 2.58 | 1.60 | 1.54 | 0.94 | $1.53 \pm 0.08$ | $1.30 \pm 0.06$ | 0.81 | 0.92 | 0.94 | 0.98 | 0.92 | 0.95 |
| | 12 | 3.19 | 1.97 | 1.81 | (N/A) | $1.81 \pm 0.12$ | $1.58 \pm 0.07$ | 0.61 | 0.88 | 0.91 | (N/A) | 0.89 | 0.92 |
| | 18 | 3.58 | 2.26 | 2.11 | (N/A) | $1.96 \pm 0.16$ | $1.79 \pm 0.08$ | 0.46 | 0.83 | 0.86 | (N/A) | 0.88 | 0.90 |
| | 24 | 4.07 | 2.48 | 2.39 | 1.33 | $2.04 \pm 0.10$ | $2.01 \pm 0.09$ | 0.35 | 0.80 | 0.83 | 0.97 | 0.86 | 0.88 |
| | 36 | 4.52 | 2.98 | 2.95 | (N/A) | $2.29 \pm 0.24$ | $2.40 \pm 0.13$ | 0.29 | 0.69 | 0.75 | (N/A) | 0.83 | 0.82 |

# E    USE OF LARGE LANGUAGE MODELS (LLMs)

We used large language model (LLM) services such as ChatGPT only to polish small parts of the paper in terms of language and readability. LLMs were not used for research ideation, methodology design, experimental implementation, data analysis, or the creation of figures and tables. All technical content, results, and conclusions are solely the work of the authors. The authors take full responsibility for the final manuscript.

