# OpenReview forum: "PIANO: Physics-Informed Autoregressive Networks"
_ICLR.cc/2026/Conference — ICLR 2026 Conference Withdrawn Submission_

### Official Review · Reviewer_xyVc · 2025-10-23

**Soundness:** 3
**Presentation:** 3
**Contribution:** 1
**Rating:** 2
**Confidence:** 4

**Summary:**

This paper introduces an autoregressive physics-informed neural network for solving time-dependent PDEs. The core contribution is conditioning predictions at time t_n on the model's own prediction at t_{n-1}, rather than treating all time points independently as in standard PINNs. The authors provide theoretical analysis of error propagation  and demonstrate improved accuracy on four canonical 1D PDEs and weather forecasting.

**Strengths:**

1- The paper correctly identifies a fundamental limitation in standard PINNs: treating all temporal collocation points symmetrically violates the causal structure of dynamical systems. This problem is well-documented, and the autoregressive formulation directly addresses it.

2- The architectural design is clear and well-motivated. The state-space formulation with learnable matrices is appropriate for temporal propagation.

3- The Physics-Informed Experience Learning paradigm elegantly combines autoregressive rollout with physics constraints. The energy formulation aggregating residuals over trajectories rather than individual points is conceptually sound.

4- The ablation study is well-designed, systematically varying finite difference order and architectural backbone across independent runs with reported standard deviations. This demonstrates that both autoregressive structure and higher-order derivatives are important.

**Weaknesses:**

1- The canonical PDE evaluation is limited to four 1D problems. While the paper demonstrates applicability to the higher-dimensional ERA5 weather forecasting task, methodological inconsistencies (teacher forcing used for weather but not canonical PDEs) and insufficient technical detail (how architecture adapts to 2D spatial domains, how finite differences work on spherical coordinates) make it difficult to assess whether the same method is being evaluated. The paper would be strengthened by evaluating established 2D PDE benchmarks (PDEBench Navier-Stokes, shallow water) using the same methodology as 1D canonical PDEs.

2- Table 1 baseline results: Were hyperparameters re-tuned for fair comparison, or were published hyperparameters used directly? PINNsFormer and PINNMamba were designed for different problems; their hyperparameters may not be optimal for Wave/Reaction/Convection/Heat equations. Appendix states "baselines rely on official implementations with reported hyperparameters," but does not clarify if any tuning was performed.

3- Critical baselines are missing. The paper cites but does not compare against directly relevant prior work:

- Wang et al. (2024) Cited three times but never compared experimentally.
- Li et al. (2024) "Causality-enhanced PINNs" Cited but not compared.
- No neural operator baselines: Fourier Neural Operator (Li et al., 2021), DeepONet (Lu et al., 2021)
- No autoregressive neural PDE baselines: PDE-Refiner (Lippe et al., 2023)


4 - Theorem 3.4 presents an error bound ||e_{n+1}|| ≤ L_G·||e_n|| + δ_n, claiming this reveals "a critical flaw in non-autoregressive PINNs." However, this error propagation form is standard in numerical analysis of time-stepping methods
The observation that standard PINNs leave δ_n unconstrained is valid, but framing standard error analysis as a novel theoretical contribution is misleading. Same for Appendix B1. The paper should clarify that it applies classical theory to motivate architectural choice, not develop a new theory.

5 - The Lipschitz continuity assumption is restrictive. It fails for:

- Shock formation in conservation laws (Burgers', Euler equations)
- Blow-up phenomena in nonlinear PDEs
- Singularities in free boundary problems
- Turbulent flows with sensitive dependence on initial conditions

All tested PDEs have smooth solutions where Lipschitz conditions hold. The theoretical framework's applicability to realistic problems with non-smooth solutions is unaddressed. The paper should discuss these limitations explicitly.

6 - Computational cost analysis is absent. No comparison of:

- Training wall-clock time
- Memory requirements during backpropagation
- Inference time per prediction step
- Scalability: how costs grow with longer horizon, higher spatial resolution, higher dimensions,

Appendix C.3 mentions GPU memory. Without cost analysis, practical utility is unclear.

7. Insufficient detail is provided on how PIANO was adapted for weather forecasting (grid resolution, initial condition handling, training procedure modifications). The comparison appears to demonstrate PIANO can be applied to ClimODE's problem formulation, not that it outperforms appropriate weather forecasting baselines
The paper states PIANO uses "autoregressive training strategy with teacher forcing", but earlier claims that avoiding teacher forcing is an advantage. This contradiction needs resolution.

**Questions:**

1. You state that PIANO does not need teacher forcing for 1D PDE, but you use it for weather forecasting. Why the difference?
2. Baselines use 'official implementations with reported hyperparameters.' Were these hyperparameters re-tuned for your specific PDEs, or used directly from the original papers (which evaluated different problems)?"

---

### Official Review · Reviewer_yYyE · 2025-10-26

**Soundness:** 3
**Presentation:** 3
**Contribution:** 3
**Rating:** 6
**Confidence:** 3

**Summary:**

The paper presents a framework aimed at improving physics-informed neural networks for time-dependent partial differential equations. It analyzes the sources of error in conventional PINNs and addresses them by introducing an explicit temporal dependency, where each prediction is conditioned on the previous one. Unlike standard formulations that treat spatial and temporal dimensions equivalently, the proposed approach models the solution at each time step as a function solely of the corresponding spatial location at the preceding time step. The temporal residual is approximated using second-order derivatives to enhance temporal accuracy. Experimental results show that this autoregressive formulation achieves better performance than classical, non-autoregressive PINNs.

**Strengths:**

- The proposed framework demonstrates consistent and significant improvements in prediction accuracy over conventional PINNs.
- The method introduces a meaningful relaxation of the original formulation through derivative approximations, enhancing flexibility both in the temporal modeling structure and in the spatial loss design.

**Weaknesses:**

- The computation of spatial derivatives appears to bypass automatic differentiation which differs to the original PINNs. This design choice should be discussed in greater depth, as it may influence training stability, computational efficiency, and accuracy. In particular, the paper should clarify how the collocation points are selected or how to select them for a certain equation and how their number and distribution affect the resulting approximation quality.
- It is unclear whether the collocation points used for training are reused during evaluation, or whether a distinct set is employed. This distinction is important for understanding the generalization ability of the proposed method to other locations in space and time.
- The paper overlooks a closely related line of work in the neural operator literature, specifically the physics-informed DeepONet framework (Wang and Perdikaris, 2023), which also introduces an autoregressive formulation for time-dependent PDEs by encoding previous-step predictions in the branch network. As the underlying idea and objective appear similar, a direct discussion or experimental comparison with this approach is important to establish the novelty and contribution of the proposed method in a broader context than PINNs.


Reference:
- Wang and Perdikaris (2023), Long-time integration of parametric evolution equations with physics-informed DeepONets (JCP)

**Questions:**

- Given the autoregressive nature of the framework, it would be helpful to evaluate how the model behaves beyond the training horizon, when applied iteratively for longer time spans.

---

### Official Review · Reviewer_jyFJ · 2025-10-29

**Soundness:** 2
**Presentation:** 2
**Contribution:** 2
**Rating:** 2
**Confidence:** 4

**Summary:**

This paper introduces PIANO, a physics-informed autoregressive network that solves time-dependent PDEs by explicitly conditioning future predictions on past states, addressing the temporal instability of standard PINNs. The authors provide a theoretical analysis proving PINNs' error propagation issue and demonstrate that PIANO's autoregressive design, trained with self-supervised rollouts, ensures stability and curbs error growth. Extensive experiments on PDE benchmarks and weather forecasting show PIANO achieves state-of-the-art accuracy and significantly outperforms existing methods.

**Strengths:**

1. It provides a rigorous theoretical analysis demonstrating the autoregressive design's superiority in controlling temporal error growth compared to standard PINNs.

2. It introduces a novel physics-informed autoregressive framework (PIANO) that achieves state-of-the-art accuracy across diverse PDE benchmarks and real-world weather forecasting.

**Weaknesses:**

1. While PIANO demonstrates improvements over standard PINNs in some aspects, it is important to note that these gains are accompanied by a significant loss of several beneficial properties that were central to the PINN methodology.

2. Since PIANO inherently employs finite difference methods for its solutions, using ground truth data that is also generated by finite difference simulations constitutes a methodological concern, as it undermines the validity of the experimental results.

**Questions:**

1. Theorem 3.3 and Theorem 4.1 both aim to demonstrate that PIANO achieves superior error control. However, to the best of my knowledge, the iterative forecasting in PIANO may instead introduce error accumulation—a phenomenon that could be mitigated by a well-trained PINN.

2. The temporal and spatial discretization employed in PIANO fundamentally undermines the advantageous properties of PINNs for super-resolution prediction.

3. The inclusion of $\hat{u}$ from the previous time step also influences the current $\hat{u}$, thereby compromising the inherent advantage of automatic differentiation in PINNs.

4. Although Theorem 4.1 provides an upper bound for error analysis, the prohibitive computational cost associated with traditional methods as $\Delta t$ and $h$ approach zero is one of the issues that the adoption of Physics-Informed Neural Networks (PINNs) aims to avoid.

5. See weaknesses.

---

### Official Review · Reviewer_5euy · 2025-10-30

**Soundness:** 3
**Presentation:** 2
**Contribution:** 2
**Rating:** 2
**Confidence:** 3

**Summary:**

The authors introduce PIANO a framework for PINNs that is trained and performs inference auto-regressively instead of one-shot prediction common in the PINN literature. The authors show that PINNs and unstable for dynamical systems modeling and PIANO curbs error growth. The authors validate this improvement through experiments on various datasets.

**Strengths:**

- Good results compared to past works on different types of dynamical systems.
- Theoretical proof on single step error bounds in PIANO.

**Weaknesses:**

- There is a lack of novelty in this work. There exists prior works that perform Auto-regressive prediction in PINNs [1,2]
- There is no discussion about the training horizon using BPTT and the testing horizon. What happens if the prediction horizon is much larger than the training one?
- What is the increase in inference time due to the autoregressive nature compared to other methods?
- Does the autoregressive paradigm introduce more errors if the behaviour of the dynamical system qualitively changes (for example a chaotic system)
- What happens if the SSM in the network is replaced with softmax attention?

[1] https://arxiv.org/pdf/2502.04018

[2] https://arxiv.org/pdf/2004.06243

**Questions:**

Please see weaknesses section.

---

### Official Review · Reviewer_d19m · 2025-11-01

**Soundness:** 2
**Presentation:** 3
**Contribution:** 2
**Rating:** 4
**Confidence:** 4

**Summary:**

The paper introduces a physics-informed autoregressive network (PIANO) for solving time-dependent PDEs by conditioning future predictions on previously computed states. The approach reformulates PINNs to include autoregression, aiming to improve temporal stability and accuracy. The authors present theoretical analysis suggesting that standard PINNs are temporally unstable and validate their method on canonical PDE benchmarks and a weather forecasting task. The results indicate performance gains compared to existing non-autoregressive PINN variants.

**Strengths:**

1. The paper addresses the issue of temporal instability in PINNs by proposing a physics-informed autoregressive formulation.

2. It presents both theoretical analysis and empirical demonstrations across PDE benchmarks and real-world weather forecasting.

3. The proposed architecture can potentially be integrated into various PINN setups.

**Weaknesses:**

1. The literature review does not adequately position PIANO among previous autoregressive PINN variants. Several related works using similar recurrent or sequence-based formulations are missing, which weakens the novelty claim.

2. Comparisons are mostly against non-autoregressive baselines. Including autoregressive approaches such as [1, 2, 3, 4, 5] would provide a fairer benchmark.

3. While the PDE benchmarks are relevant, the method has not been validated on more complex, nonlinear, or higher-order PDEs, which makes it difficult to ascertain its robustness and generality.

4. It is unclear whether the proposed model can handle irregular or non-Cartesian geometries, as all experiments appear to be conducted on fixed, uniform grids.

5. The connection of PIANO to neural ODE and recurrent neural PDE solvers is evident, but this similarity is not discussed, giving the impression of overlap rather than a clear methodological distinction.

6. The paper is difficult to read, as the authors introduce multiple terminologies (PIEL, PIANO), which can be confusing.

7. The paper does not clearly specify what exactly is minimized, whether it is energy, the PDE residual, or another form of physics-constrained objective, and how boundary and initial conditions are incorporated.

8. Additional clarification on the autoregressive loss design, gradient stability over long rollouts, and computational scaling would help substantiate the claimed advantages.

[1] Lippe, Phillip, et al. "Pde-refiner: Achieving accurate long rollouts with neural pde solvers." Advances in Neural Information Processing Systems 36 (2023): 67398-67433.

[2] Kapoor, Taniya, et al. "Neural oscillators for generalization of physics-informed machine learning." Proceedings of the AAAI Conference on Artificial Intelligence. Vol. 38. No. 12. 2024.

[3] Bergamin, Federico, et al. "Guided autoregressive diffusion models with applications to PDE simulation." ICLR 2024 Workshop on AI4DifferentialEquations In Science. 2024.

[4] Michałowska, Katarzyna, et al. "Neural operator learning for long-time integration in dynamical systems with recurrent neural networks." 2024 International Joint Conference on Neural Networks (IJCNN). IEEE, 2024.

[5] Koehler, Felix, Simon Niedermayr, and Nils Thuerey. "APEBench: A benchmark for autoregressive neural emulators of PDEs." Advances in Neural Information Processing Systems 37 (2024): 120252-120310.

**Questions:**

1. How does PIANO differ fundamentally from existing autoregressive or recurrent PINN formulations such as those in [1, 2, 3, 4, 5]?

2. Comparisons against other autoregressive physics-informed frameworks, rather than only non-autoregressive baselines, would be beneficial.

3. How does the model perform on strongly nonlinear or chaotic PDEs?

4. Is the method limited to uniform Cartesian grids, or can it be generalized to irregular domains and complex geometries?

5. It would be beneficial to clarify how boundary and initial conditions are integrated into the loss, and why certain terms (such as the initial condition) appear to be omitted.

6. What is the computational overhead of the autoregressive rollout compared to traditional PINNs or neural operators?

7. Given the similarities with neural ODEs or recurrent sequence models, please discuss the conceptual differences that can improve temporal stability.

8. Has the impact of gradient accumulation over time on training stability and efficiency been examined for long sequences?

---

### Author Response · Authors · 2025-12-04
**Clarifying misunderstandings in the reviews.**

Thank you to the reviewers for their review. Several key concerns in the reviews arise from conflating our setting with data-driven operator learning, and from reading coordinate-sequential architectures as already autoregressive. We summarise the main clarifications briefly for the area chairs.

1. **Novelty and scope**
PIANO is, to our knowledge, the first physics-informed *autoregressive* framework that is trained fully self-supervised using only PDE residuals and BC/IC information, with no trajectory pairs $(u_t \to u_{t+1})$. It learns a state-transition map
$$\hat u(x, t_n) = f_\theta\big(x, t_n, \hat u(x, t_{n-1})\big)$$
and is optimized on rollout-based residual energies that explicitly constrain the one-step defect $\delta_n$ (Theorems 3.3 and 4.1). Coordinate-sequential PINN variants (PINNsFormer, PINNMamba, causal weighting, curricula) still implement $u(x,t_n) = f_\theta(x,t_n)$ and do not propagate state in time. Physics-informed DeepONet and PDE-Refiner are trained on supervised trajectories and do not provide an unsupervised BPTT rollout with residual control.

2. **Comparisons to neural operators and autoregressive operator learners**
The PDE benchmarks are standard single-instance PINN tasks: one solves a given PDE from its equations, with no dataset of trajectories. Neural operators (FNO, DeepONet, PDE-Refiner, etc.) require many solution trajectories and are therefore not applicable in this regime in a scientifically meaningful way. On these benchmarks we therefore compare against a broad set of PINN-style baselines, including the strongest sequential models (PINNsFormer, PINNMamba), and PIANO attains the best performance across all equations (Table 1). Where trajectory data is available, namely ERA5, we do compare directly to neural-operator-style methods (FourCastNet, ClimaX) and to ClimODE, and PIANO improves RMSE/ACC across all variables and lead times (Figure 6).

3. **Use of finite differences and benchmark choice**
Our use of second-order finite differences to evaluate residuals on the predicted grid follows common practice in time-dependent PINNs and is explicitly accounted for in the consistency term $\kappa(\Delta t^p + h^q)$ in Theorem 4.1. The ground truth is a higher-resolution numerical solution, which is the standard evaluation protocol in this literature. The PDE set (Wave, Reaction, Convection, Heat) is exactly the suite used by recent PINNsFormer, PINNMamba and RoPINNs works, chosen for methodological comparability rather than to claim coverage of all non-smooth or chaotic regimes.

4. **Temporal stability and cost**
Our theory shows that non-autoregressive PINNs leave the rollout error $\delta_n$ unconstrained, while PIANO directly ties $\delta_n$ to the rollout residual energy, leading to near-flat error growth in time (Figure 4), unlike the rapidly growing errors of PINNsFormer and PINNMamba. Appendix C.3 reports that the training and inference costs of PIANO are comparable to these sequential baselines, often lower due to the lightweight state-space backbone.

In summary, PIANO is not a minor architectural tweak, but a physics-informed autoregressive reformulation of PINNs with explicit theoretical and empirical control of temporal error growth, state-of-the-art results on standard PDE benchmarks, and competitive performance against neural operators in the only setting where such a comparison is appropriate (ERA5).

---

### Note · Authors · 2026-01-19

I have read and agree with the venue's withdrawal policy on behalf of myself and my co-authors.